# From Complexity to Simplicity: Adaptive ES-Active Subspaces for Blackbox Optimization

**Krzysztof Choromanski**[*]
Google Brain Robotics
kchoro@google.com

**Aldo Pacchiano**[*]
UC Berkeley
pacchiano@berkeley.edu

**Jack Parker-Holder**[*]
University of Oxford
jackph@robots.ox.ac.uk

**Yunhao Tang**[*]
Columbia University
yt2541@columbia.edu

**Vikas Sindhwani**
Google Brain Robotics
sindhwani@google.com

## Abstract

We present a new algorithm (ASEBO) for optimizing high-dimensional blackbox functions. ASEBO adapts to the geometry of the function and learns optimal sets of sensing directions, which are used to probe it, on-the-fly. It addresses the exploration-exploitation trade-off of blackbox optimization with expensive blackbox queries by continuously learning the bias of the lower-dimensional model used to approximate gradients of smoothings of the function via compressed sensing and contextual bandits methods. To obtain this model, it leverages techniques from the emerging theory of active subspaces [8] in a novel ES blackbox optimization context. As a result, ASEBO learns the dynamically changing intrinsic dimensionality of the gradient space and adapts to the hardness of different stages of the optimization without external supervision. Consequently, it leads to more sample-efficient blackbox optimization than state-of-the-art algorithms. We provide theoretical results and test ASEBO advantages over other methods empirically by evaluating it on the set of reinforcement learning policy optimization tasks as well as functions from the recently open-sourced Nevergrad library.

## 1 Introduction

Consider a high-dimensional function $F : \mathbb{R}^d \rightarrow \mathbb{R}$. We assume that querying it is expensive. Examples include reinforcement learning (RL) blackbox functions taking as inputs vectors $\theta$ encoding policies $\pi : \mathcal{S} \rightarrow \mathcal{A}$ mapping states to actions and outputting total (expected/discounted) rewards obtained by agents applying $\pi$ in given environments [6]. For this class of functions evaluations usually require running a simulator. Other examples include wind configuration design optimization problems for high speed civil transport aircrafts, optimizing computer codes (e.g. NASA synthetic tool FLOPS/ENGENN used to size the aircraft and propulsion system [2]), crash tests, medical and chemical reaction experiments [37].

Evolution strategy (ES) methods have traditionally been used in low-dimensional regimes (e.g. hyperparameter tuning), and considered ill-equipped for higher dimensional problems due to poor sampling complexity [27]. However, a flurry of recent work has shown they can scale better than previously believed [33, 11, 29, 25, 7, 30, 21]. This is thanks to a couple of reasons.

First of all, new ES methods apply several efficient heuristics (filtering, various normalization techniques as in [25] and new exploration strategies as in [11]) in order to substantially improve

---

[*]Equal contribution.

sampling complexity. Other recent methods [29, 7] are based on more accurate Quasi Monte Carlo (MC) estimators of the gradients of Gaussian smoothings of blackbox functions with theoretical guarantees. These approaches provide better quality gradient sensing mechanisms. Additionally, in applications such as RL, new compact structured policy architectures (such as low-displacement rank neural networks from [7] or even linear policies [14]) are used to reduce the number of policies' parameters and dimensionality of the optimization problem.

Recent research also shows that ES-type blackbox optimization in RL leads to more stable policies than policy gradient methods since ES methods search for parameters that are robust to perturbations [19]. Unlike policy gradient methods, ES aims to find parameters maximizing expected reward (rather than just a reward) in respect to Gaussian perturbations.

Finally, pure ES methods as opposed to state-of-the-art policy optimization techniques (TRPO, PPO or ARS [32, 15, 31, 25]), can be applied also for blackbox optimization problems that do not exhibit MDP structure required for policy gradient methods and cannot benefit from state normalization algorithm central to ARS. This has led to their recent popularity for non-differentiable tasks [17, 14].

In this paper we introduce a new adaptive sample-efficient blackbox optimization algorithm. ASEBO adapts to the geometry of blackbox functions and learns optimal sets of sensing directions, which are used to probe them, on-the-fly. To do this, it leverages techniques from the emerging theory of active subspaces [8, 10, 9, 20] in a novel ES blackbox optimization context. Active subspaces and their extensions are becoming popular as effective techniques for dimensionality reduction (see for instance: active manifolds [5] or ResNets for learning isosurfaces [36]). However, to the best of our knowledge we are the first to apply active subspace ideas for ES optimization.

ASEBO addresses the exploration-exploitation trade-off of blackbox optimization with expensive function queries by continuously learning the bias of the lower-dimensional model used to approximate gradients of smoothings of the function with compressed sensing and contextual bandits methods. The adaptiveness is what distinguishes it from some recently introduced guided ES methods such as [24] that rely on fixed hyperparameters that are hard to tune in advance (e.g. the length of the buffer defining lower dimensional space for gradient search). We provide theoretical results and empirically evaluate ASEBO on a set of RL blackbox optimization tasks as well as non-RL blackbox functions from the recently open-sourced Nevergrad library [34], showing that it consistently learns optimal inputs with fewer queries to a blackbox function than other methods.

ASEBO **versus CMA-ES:** There have been a variety of works seeking to reduce sampling complexity for ES methods through the use of metric learning. The prominent class of the covariance matrix adaptation evolution strategy (CMA-ES) methods derives state-of-the-art derivative free blackbox optimization algorithms, which seek to learn and maintain a fully parameterized Gaussian distribution. CMA-ES suffers from quadratic time complexity for each evaluation which can be limiting for high dimensional problems. As such, a series of attempts have been made to produce scalable variants of CMA-ES, by restricting the covariance matrix to the diagonal (sep-CMA-ES [28]) or a low rank approximation as in VD-CMA-ES [3] and LM-CMA-ES [22]. Two recent algorithms, VkD-CMA-ES [4] and LM-MA-ES [23], seek to combine the above ideas and have been shown to be successful in large-scale settings, including RL policy learning [26]. Although these methods are able to quickly learn and adapt the covariance matrix, they are heavily dependent on hyperparameter selection [4, 35] and lack the means to avoid learning a bias. As our experiments show, this can severely hurt their performance. The best CMA-ES variants often struggle with RL tasks of challenging objective landscapes, displaying inconsistent performance across tasks. Furthermore, they require careful hyperparameter tuning for good performance (see: analysis in Section 4, Fig. 3).

## 2  Adaptive Sample-Efficient Blackbox Optimization

Before we describe ASEBO, we explain key theoretical ideas behind the algorithm. ASEBO uses online PCA to maintain and update on-the-fly subspaces which we call *ES-active subspaces* $\mathcal{L}_{\text{active}}^{\text{ES}}$, accurately approximating the gradient data space at a given phase of the algorithm. The bias of the obtained gradient estimators is measured by sensing the length of its component from the orthogonal complement $\mathcal{L}_{\text{active}}^{\text{ES},\perp}$ via compressed sensing or computing optimal probabilities for exploration (e.g. sensing from $\mathcal{L}_{\text{active}}^{\text{ES},\perp}$) via contextual bandits methods [1]. The algorithm corrects its probabilistic distributions used for choosing directions for gradient sensing based on these measurements. As we show, we can measure that bias accurately using only a constant number of additional function

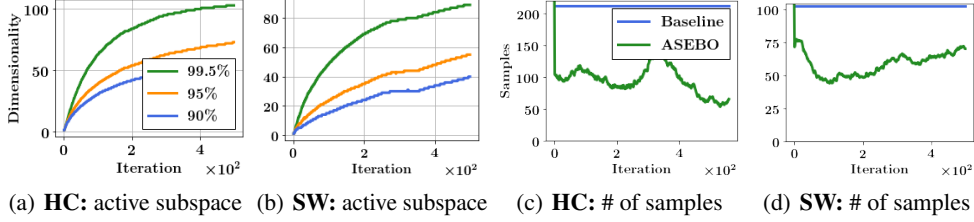

(a) **HC:** active subspace    (b) **SW:** active subspace    (c) **HC:** # of samples    (d) **SW:** # of samples

Figure 1: The motivation behind ASEBO. Two first plots: ES baseline for HalfCheetah and Swimmer tasks from the OpenAI Gym library for 212-dimensional policies - the plot shows how the dimensionality of the space capturing a given percentage of variance of approximate gradient data depends on the iteration of the algorithm. This information is never exploited by the algorithm, even though 99.5% of the variance resides in the much lower-dimensional space (100 dimensions). Two last plots: ASEBO taking advantage of this information (# of sample/sensing directions reflects the hardness of the optimization at each iteration and is strongly correlated with the PCA dimensionality.

queries, regardless of the dimensionality. This in turn determines an exploration strategy, as we explain later. Estimated gradients are then used to update parameters.

## 2.1 Preliminaries

Consider a blackbox function $F : \mathbb{R}^d \to \mathbb{R}$. We do not assume that $F$ is differentiable. The *Gaussian smoothing* [27] $F_\sigma$ of $F$ parameterized by smoothing parameter $\sigma > 0$ is given as: $F_\sigma(\theta) = \mathbb{E}_{\mathbf{g} \in \mathcal{N}(0, \mathbf{I}_d)}[F(\theta + \sigma\mathbf{g})] = (2\pi)^{-\frac{d}{2}} \int_{\mathbb{R}^d} F(\theta + \sigma\mathbf{g})e^{-\frac{\|\mathbf{g}\|^2}{2}} d\mathbf{g}$. The gradient of the Gaussian smoothing of $F$ is given by the formula:

$$\nabla F_\sigma(\theta) = \frac{1}{\sigma}\mathbb{E}_{\mathbf{g} \sim \mathcal{N}(0, \mathbf{I}_d)}[F(\theta + \sigma\mathbf{g})\mathbf{g}]. \tag{1}$$

Formula 1 on $\nabla F_\sigma(\theta)$ leads straightforwardly to several unbiased Monte Carlo (MC) estimators of $\nabla F_\sigma(\theta)$, where the most popular ones are: the *forward finite difference* estimator [7] defined as: $\widehat{\nabla}_{\mathrm{MC}}^{\mathrm{FD}} F_\sigma(\theta) = \frac{1}{k\sigma} \sum_{i=1}^k (F(\theta + \sigma\mathbf{g}_i) - F(\theta))\mathbf{g}_i$, and an *antithetic ES gradient estimator* [30] given as: $\widehat{\nabla}_{\mathrm{MC}}^{\mathrm{AT}} F_\sigma(\theta) = \frac{1}{2k\sigma} \sum_{i=1}^k (F(\theta + \sigma\mathbf{g}_i) - F(\theta - \sigma\mathbf{g}_i))\mathbf{g}_i$, where typically $\mathbf{g}_1, ..., \mathbf{g}_k$ are taken independently at random from $\mathcal{N}(0, \mathbf{I}_d)$ of from more complex joint distributions for variance reduction (see: [7]). We call samples $\mathbf{g}_1, ..., \mathbf{g}_k$ the *sensing directions* since they are used to sense gradients $\nabla F_\sigma(\theta)$. The antithetic formula can be alternatively rationalized as giving the renormalized gradient of $F$ (if $F$ is smooth), if not taking into account cubic and higher-order terms of the Taylor expansion $F(\theta + \mathbf{v}) = F(\theta) + \nabla F^\top \mathbf{v} + \frac{1}{2}\mathbf{v}^\top H(\theta)\mathbf{v}$ (where $H(\theta)$ stands for the Hessian of $F$ in $\theta$).

Standard ES methods apply different gradient-based techniques such as SGD or Adam, fed with the above MC estimators of $\nabla F_\sigma$ to conduct blackbox optimization. The number of samples $k$ per iteration of the optimization procedure is usually of the order $O(d)$. This becomes a computational bottleneck for high-dimensional blackbox functions $F$ (for instance, even for relatively small RL tasks with policies encoded by compact neural networks we still have $d > 100$ parameters).

## 2.2 ES-active subspaces via online PCA with decaying weights

The first idea leading to the ASEBO algorithm is that in practice one does not need to estimate the gradient of $F$ accurately (after all ES-type methods do not even aim to compute the gradient of $F$, but rather focus on $\nabla F_\sigma$). Poor scalability of ES-type blackbox optimization algorithms is caused by high-dimensionality of the gradient vector. However, during the optimization process the space spanned by gradients may be locally well approximated by a lower-dimensional subspace $\mathcal{L}$ and sensing the gradient in that subspace might be more effective. In some recent papers such as [24] such a subspace is defined simply as $\mathcal{L} = \mathrm{span}\{\widehat{\nabla}_{\mathrm{MC}}^{\mathrm{AT}} F_\sigma(\theta_i), \widehat{\nabla}_{\mathrm{MC}}^{\mathrm{AT}} F_\sigma(\theta_{i-1}), ..., \widehat{\nabla}_{\mathrm{MC}}^{\mathrm{AT}} F_\sigma(\theta_{i-s+1})\}$, where $\{\widehat{\nabla}_{\mathrm{MC}}^{\mathrm{AT}} F_\sigma(\theta_i), \widehat{\nabla}_{\mathrm{MC}}^{\mathrm{AT}} F_\sigma(\theta_{i-1}), ..., \widehat{\nabla}_{\mathrm{MC}}^{\mathrm{AT}} F_\sigma(\theta_{i-s+1})\}$ stands for the batch of last $s$ approximated gradients during the optimization process and $s$ is a fixed hyperparameter. Even though $\mathcal{L}$ will dynamically change during the optimization, such an approach has several disadvantages in practice. Tuning parameter $s$ is very difficult or almost impossible and the assumption that the dimensionality of $\mathcal{L}$ should be constant during optimization is usually false. In our approach, dimensionality of $\mathcal{L}$ varies and depends on the hardness of the optimization in different optimization stages.

We apply Principal Component Analysis (PCA, [18]) to create a subspace $\mathcal{L}$ capturing particular variance $\sigma > 0$ of the approximate gradients data. This data is either: the approximate gradients computed in previous iterations from the antithetic formula or: the elements of the sum from that equation that are averaged over to obtain these gradients. For clarity of the exposition, from now on we will assume the former, but both variants are valid. Choosing $\sigma$ is in practice much easier than $s$ and leads to subspaces $\mathcal{L}$ of varying dimensionalities throughout the optimization procedure, called by us from now on *ES-active subspaces* $\mathcal{L}_{\text{active}}^{\text{ES}}$.

---

**Algorithm 1** ASEBO Algorithm

---

**Hyperparameters:** number of iterations of full sampling $l$, smoothing parameter $\sigma > 0$, step size $\eta$, PCA threshold $\epsilon$, decay rate $\gamma$, total number of iterations $T$.
**Input:** blackbox function $F$, vector $\theta_0 \in \mathbb{R}^d$ where optimization starts. $\text{Cov}_0 \in \{0\}^{d \times d}$, $p^0 = 0$.
**Output:** vector $\theta_T$.
**for** $t = 0, \ldots, T - 1$ **do**
    **if** $t < l$ **then**
        Take $n_t = d$. Sample $\mathbf{g}_1, \cdots, \mathbf{g}_{n_t}$ from $\mathcal{N}(0, \mathbf{I}_d)$ (independently).
    **else**
        1. Take top $r$ eigenvalues $\lambda_i$ of $\text{Cov}_t$, where $r$ is smallest such that: $\sum_{i=1}^r \lambda_i \geq \epsilon \sum_{i=1}^d \lambda_i$, using its SVD as described in text and take $n_t = r$.
        2. Take the corresponding eigenvectors $\mathbf{u}_1, ..., \mathbf{u}_r \in \mathbb{R}^d$ and let $\mathbf{U} \in \mathbb{R}^{d \times r}$ be obtained by stacking them together. Let $\mathbf{U}^{\text{act}} \in \mathbb{R}^{d \times r}$ be obtained from stacking together some orthonormal basis of $\mathcal{L}_{\text{active}}^{\text{ES}} \overset{\text{def}}{=} \text{span}\{\mathbf{u}_1, ..., \mathbf{u}_r\}$. Let $\mathbf{U}^\perp \in \mathbb{R}^{d \times (d-r)}$ be obtained from stacking together some orthonormal basis of the orthogonal complement $\mathcal{L}_{\text{active}}^{\text{ES},\perp}$ of $\mathcal{L}_{\text{active}}^{\text{ES}}$.
        3. Sample $n_t$ vectors $\mathbf{g}_1, ..., \mathbf{g}_{n_t}$ as follows: with probability $1 - p^t$ from $\mathcal{N}(0, \mathbf{U}^\perp (\mathbf{U}^\perp)^\top)$ and with probability $p^t$ from $\mathcal{N}(0, \mathbf{U}^{\text{act}} (\mathbf{U}^{\text{act}})^\top)$.
        4. Renormalize $\mathbf{g}_1, ..., \mathbf{g}_{n_t}$ such that marginal distributions $\|\mathbf{g}_i\|_2$ are $\chi(d)$.
    1. Compute $\widehat{\nabla}_{\text{MC}}^{\text{AT}} F(\theta_t)$ as: $\widehat{\nabla}_{\text{MC}}^{\text{AT}} F(\theta_t) = \frac{1}{2 n_t \sigma} \sum_{j=1}^{n_t} (F(\theta_t + \mathbf{g}_j) - F(\theta_t - \mathbf{g}_j)) \mathbf{g}_j$.
    2. Set $\text{Cov}_{t+1} = \lambda \text{Cov}_t + (1 - \lambda) \Gamma$, where $\Gamma = \widehat{\nabla}_{\text{MC}}^{\text{AT}} F_\sigma(\theta_t) (\widehat{\nabla}_{\text{MC}}^{\text{AT}} F_\sigma(\theta_t))^\top$.
    3. Set $p^{t+1} = p_{\text{opt}}$ for $p_{\text{opt}}$ output by Algorithm 2 and: $\theta_{t+1} \leftarrow \theta_t + \eta \widehat{\nabla}_{\text{MC}}^{\text{AT}} F(\theta_t)$.

---

These will be in turn applied to define covariance matrices encoding probabilistic distributions applied to construct sensing directions used for estimating $\nabla F_\sigma(\theta)$. The additional advantage of our approach is that PCA automatically filters out gradient noise.

We use our own online version of PCA with decaying weights (decay rate is defined by parameter $\lambda > 0$). By tuning $\lambda$ we can define the rate at which historical approximate gradient data is used to choose the right sensing directions, which will continuously decay. We consider a stream of approximate gradients $\widehat{\nabla}_{\text{MC}}^{\text{AT}} F_\sigma(\theta_0), ... \widehat{\nabla}_{\text{MC}}^{\text{AT}} F_\sigma(\theta_i), ...$ obtained during the optimization procedure. We maintain and update on-the-fly the covariance matrix $\text{Cov}_t$, where $t$ stands for the number of completed iterations, in the form of the symmetric matrix SVD-decomposition $\text{Cov}_t = \mathbf{Q}_t^\top \Sigma_t \mathbf{Q}_t \in \mathbb{R}^d$. When the new approximate gradient $\widehat{\nabla}_{\text{MC}}^{\text{AT}} F_\sigma(\theta_t)$ arrives, the update of the covariance matrix is driven by the following equation, reflecting data decay process, where $\mathbf{x}_t = \widehat{\nabla}_{\text{MC}}^{\text{AT}} F_\sigma(\theta_t)$:

$$\mathbf{Q}_{t+1}^\top \Sigma_{t+1} \mathbf{Q}_{t+1} = \lambda \mathbf{Q}_t^\top \Sigma_t \mathbf{Q}_t + (1 - \lambda) \mathbf{x}_t \mathbf{x}_t^\top, \tag{2}$$

To conduct the update cheaply, it suffices to observe that the RHS of Equation 2 can be rewritten as: $\lambda \mathbf{Q}_t^\top \Sigma_t \mathbf{Q}_t + (1 - \lambda) \mathbf{x}_t \mathbf{x}_t^\top = \mathbf{Q}_t^\top (\lambda \Sigma_t + (1 - \lambda) \mathbf{Q}_t \mathbf{x}_t (\mathbf{Q}_t \mathbf{x}_t)^\top) \mathbf{Q}_t$. Now, using the fact that for a matrix of the form $\mathbf{D} + \mathbf{u} \mathbf{u}^\top$, we can get its decomposition in time $O(d^2)$ [13], we obtain an algorithm performing updates in quadratic time. That in practice suffices since the bottleneck of the computations is in querying $F$ and additional overhead related to updating $\mathcal{L}_{\text{active}}^{\text{ES}}$ is negligible.

**ES-active subspaces versus active subspaces:** Our mechanism for constructing $\mathcal{L}_{\text{active}}^{\text{ES}}$ is inspired by the recent theory of active subspaces [8], developed to determine the most important directions in the space of input parameters of high-dimensional blackbox functions such as computer simulations.

The *active subspace* of a differentiable function $F : \mathbb{R}^d \to \mathbb{R}$, square-integrable with respect to the given probabilistic density function $\rho : \mathbb{R}^d \to \mathbb{R}$, is given as a linear subspace $\mathcal{L}_{\text{active}}$ defined by the

first $r$ (for a fixed $r < d$) eigenvectors of the following $d \times d$ symmetric positive definite matrix:

$$\text{Cov} = \int_{\mathbf{x} \in \mathbb{R}^d} \nabla F(\mathbf{x}) \nabla F(\mathbf{x})^\top \rho(\mathbf{x}) d\mathbf{x} \tag{3}$$

Density function $\rho$ determines where compact representation of $F$ is needed. In our approach we do not assume that $\nabla F$ exists, but the key difference between $\mathcal{L}_{\text{active}}^{\text{ES}}$ and $\mathcal{L}_{\text{active}}$ lies somewhere else.

The goal of ASEBO is to avoid approximating the exact gradient $\nabla F(\mathbf{x}) \in \mathbb{R}^d$ which is what makes standard ES methods very expensive and which is done in [9] via gradient sketching techniques combined with finite difference approaches (standard methods of choice for ES baselines).

---

**Algorithm 2** Explore estimator via exponentiated sampling

---

**Hyperparameters:** smoothing parameter $\sigma$, horizon $C$, learning rate $\alpha$, probability regularizer $\beta$, initial probability parameter $q_0^t \in (0, 1)$.
**Input:** subspaces: $\mathcal{L}_{\text{active}}^{\text{ES}}, \mathcal{L}_{\text{active}}^{\text{ES},\perp}$, function $F$, vector $\theta_t$
**Output:**
**for** $l = 1, \cdots, C + 1$ **do**
    1. Compute $p_{l-1}^t = (1 - 2\beta)q_{l-1}^t + \beta$ and sample $a_l^t \sim \text{Ber}(p_l^t)$.
    3. If $a_l^t = 1$, sample $\mathbf{g}_l \sim \mathcal{N}(0, \sigma \mathbf{I}_{\mathcal{L}_{\text{active}}^{\text{ES}}})$, otherwise sample $\mathbf{g}_l \sim \mathcal{N}(0, \sigma \mathbf{I}_{\mathcal{L}_{\text{active}}^{\text{ES},\perp}})$.
    4. Compute $v_l = \frac{1}{2\sigma}\left(F(\theta_t + \mathbf{g}_l) - F(\theta_t - \mathbf{g}_l)\right)$.
    5. Set $\mathbf{e}_l = (1 - 2\beta)\begin{bmatrix} \left(-\frac{a_l^t(\dim(\mathcal{L}_{\text{active}}^{\text{ES}})+2)}{(p_l^t)^3}\right) \\ \left(-\frac{(1-a_l^t)(\dim(\mathcal{L}_{\text{active}}^{\text{ES},\perp})+2)}{(1-p_l^t)^3}\right) \end{bmatrix} v_l^2$.
    6. Set $q_l^t = \frac{q_{l-1}^t \exp(-\alpha \mathbf{e}_l(1))}{q_{l-1}^t \exp(-\alpha \mathbf{e}_l(1)) + (1 - q_{l-1}^t)\exp(-\alpha \mathbf{e}_l(2))}$.
**Return:** $p_C$.

---

Instead, in ASEBO an ES-active subspace $\mathcal{L}_{\text{active}}^{\text{ES}}$ is itself used to define sensing directions and the number of chosen samples $k$ is given by the dimensionality of $\mathcal{L}_{\text{active}}^{\text{ES}}$. This drastically reduces sampling complexity, but comes at a price of risking the optimization to be trapped in the fixed lower-dimensional space that will not be representative for gradient data as optimization progresses. We propose a solution requiring only a constant number of extra queries to $F$ in the next sections.

### 2.3 Exploration-exploitation trade-off: Adaptive Exploration Mechanism

The procedure described above needs to be accompanied with an exploration strategy that will determine how frequently to choose sensing directions outside the constructed on-the-fly lower-dimensional ES-subspace $\mathcal{L}_{\text{active}}^{\text{ES}}$. Our exploration strategies will be encoded by hybrid probabilistic distributions for sampling sensing directions. The frequency of sensing from the distributions with covariance matrices obtained from $\mathcal{L}_{\text{active}}^{\text{ES}}$ (corresponding to exploitation) and from its orthogonal complement $\mathcal{L}_{\text{active}}^{\text{ES},\perp}$ or entire space (corresponding to exploration) will be given by weights encoding the importance of exploitation versus exploration in any given iteration of the optimization. For a vector $\mathbf{x} \in \mathbb{R}^d$ denote by $\mathbf{x}_{\text{active}}$ its projection onto $\mathcal{L}_{\text{active}}^{\text{ES}}$ and by $\mathbf{x}_\perp$ its projection onto $\mathcal{L}_{\text{active}}^{\text{ES},\perp}$. The useful metric that can be used to update the above weights in an online manner in the $t^{th}$ iteration of the algorithm is the ratio: $r = \frac{\|(\nabla F_\sigma(\theta_t))_{\text{active}}\|_2}{\|(\nabla F_\sigma(\theta_t))_\perp\|_2}$. Smaller values of $r$ indicate that current active subspace is not representative enough for the gradient and more aggressive exploration needs to be conducted. In practice, we do not compute $r$ explicitly, but rather its approximated version $\widehat{r}$.

One can simply take: $\widehat{r} = \frac{\|(\widehat{\nabla}_{\text{MC}}^{\text{AT}} F_\sigma(\theta_{t-1}))_{\text{active}}\|_2}{\|(\widehat{\nabla}_{\text{MC}}^{\text{AT}} F_\sigma(\theta_{t-1}))_\perp\|_2}$, where $\widehat{\nabla}_{\text{MC}}^{\text{AT}} F_\sigma(\theta_{t-1})$ is obtained in the previous iteration. But we can do better. It suffices to separately estimate $\|(\nabla F_\sigma(\theta_t))_{\text{active}}\|_2$ and $\|(\nabla F_\sigma(\theta_t))_\perp\|_2$. However we do not aim to estimate $(\nabla F_\sigma(\theta_t))_{\text{active}}$ and $(\nabla F_\sigma(\theta_t))_\perp$. That would be equivalent to computing exact estimate of $\nabla F_\sigma(\theta_t)$, defeating the purpose of ASEBO. Instead, we note that estimating the length of the unknown high-dimensional vector is much simpler than estimating the vector itself and can be done in the probabilistic manner with arbitrary precision via the set of dot-product queries of size independent from dimensionality $d$ via compressed sensing methods. We refine this approach and propose more accurate contextual bandits method that also relies on dot-product queries applied in the ES-context, but aims to directly approximate optimal probabilities

of sampling from $\mathcal{L}^{\mathrm{ES}}_{\mathrm{active}}$ rather than approximating gradients components' lengths (see Algorithm 2 box, the compressed sensing baseline is presented in the Appendix). The related computational overhead is measured in constant number of extra function queries, negligible in practice.

## 2.4 The Algorithm

ASEBO is given in the Algorithm 1 box. The algorithm we apply to score relative importance of sampling from the ES-active subspace $\mathcal{L}^{\mathrm{ES}}_{\mathrm{active}}$ versus from outside $\mathcal{L}^{\mathrm{ES}}_{\mathrm{active}}$ is in the Algorithm 2 box.

As we have already mentioned, it uses bandits method do determine optimal probability of sampling from $\mathcal{L}^{\mathrm{ES}}_{\mathrm{active}}$. In the next section we show that by using these techniques we can substantially reduce the variance of ES blackbox gradient estimators if ES-active subspaces approximate the gradient data well (which is the case for RL blackbox functions as presented in Fig. 1). Horizon lengths $C$ in Algorithm 2 which determines the number of extra function queries should be in practice chosen as small constants. In each iteration of Algorithm 1 the number of function queries is proportional to the dimensionality of the ES-active subspace $\mathcal{L}^{\mathrm{ES}}_{\mathrm{active}}$ rather than the original space.

## 3 Theoretical Results

We provide here theoretical guarantees for the ASEBO sampling mechanism (in Algorithm 1), where sensing directions $\{\mathbf{g}_i\}$ at time $t$ are sampled from the hybrid distribution $\widehat{P}$: with probability $p^t$ from $\mathcal{N}(0, \mathbf{I}_{\mathcal{L}_{\mathrm{active}}})$ and with probability $1 - p^t$ from $\mathcal{N}(0, \mathbf{I}_{\mathcal{L}^{\perp}_{\mathrm{active}}})$.

Following notation in Algorithm 1, let $\mathbf{U}^{\mathrm{act}} \in \mathbb{R}^{d \times r}$ be obtained by stacking together vectors of some orthonormal basis of $\mathcal{L}^{\mathrm{ES}}_{\mathrm{active}}$, where $\dim(\mathcal{L}^{\mathrm{ES}}_{\mathrm{active}}) = r$ and let $\mathbf{U}^{\perp} \in \mathbb{R}^{d \times (d-r)}$ be obtained my stacking together vectors of some orthonormal basis of its orthogonal complement $\mathcal{L}^{\mathrm{ES},\perp}_{\mathrm{active}}$. Denote by $\sigma$ a smoothing parameter. We make the following regularity assumptions on $F$:

**Assumption 1.** $F$ is $L-$Lipschitz, i.e. for all $\theta, \theta' \in \mathbb{R}^d$, $|F(\theta) - F(\theta')| \leq L\|\theta - \theta'\|_2$.

**Assumption 2.** $F$ has a $\tau$-smooth third order derivative tensor with respect to $\sigma > 0$, so that $F(\theta + \sigma\mathbf{g}) = F(\theta) + \sigma\nabla F(\theta)^\top \mathbf{g} + \frac{\sigma^2}{2}\mathbf{g}^\top H(\theta)\mathbf{g} + \frac{1}{6}\sigma^3 F'''(\theta)[\mathbf{v}, \mathbf{v}, \mathbf{v}]$ for some $\mathbf{v} \in \mathbb{R}^d$ ($\|\mathbf{v}\|_2 \leq \|\mathbf{g}\|_2$) satisfying $|F'''(\theta)[\mathbf{v}, \mathbf{v}, \mathbf{v}]| \leq \tau\|\mathbf{v}\|_2^3 \leq \tau\|\mathbf{g}\|_2^3$.

Observe that: $\mathbb{E}_{\mathbf{g}\sim\widehat{P}}\left[\mathbf{g}\mathbf{g}^\top\right] = \left(p^t\mathbf{U}^{\mathrm{act}}\left(\mathbf{U}^{\mathrm{act}}\right)^\top + (1 - p^t)\mathbf{U}^{\perp}\left(\mathbf{U}^{\perp}\right)^\top\right)$. Define $C_1 = \left(p^t\mathbf{U}^{\mathrm{act}}\left(\mathbf{U}^{\mathrm{act}}\right)^\top + (1 - p^t)\mathbf{U}^{\perp}\left(\mathbf{U}^{\perp}\right)^\top\right)$. Let $\widehat{\nabla}^{\mathrm{AT,asebo}}_{\mathrm{MC},k=1}F_\sigma(\theta) = \mathbf{C}_1^{-1}\frac{F(\theta+\sigma\mathbf{g})\mathbf{g}+F(\theta+\sigma\mathbf{g})(-\mathbf{g})}{2\sigma}$ be the gradient estimator corresponding to $\widehat{P}$. We will assume that $\sigma$ is small enough, i.e. $\sigma < \frac{1}{35}\sqrt{\frac{\epsilon\min(p^t,1-p^t)}{\tau d^3\max(L,1)}}$ for some precision parameter $\epsilon > 0$. Our first result shows that under these assumptions, baseline and ASEBO estimators of $\nabla_\sigma F(\theta)$ are also good estimators of $\nabla F(\theta)$:

**Lemma 3.1.** *If $F$ satisfies Assumptions 1 and 2, the estimators $\widehat{\nabla}^{\mathrm{AT,base}}_{\mathrm{MC},k=1}F_\sigma(\theta)$ and $\widehat{\nabla}^{\mathrm{AT,asebo}}_{\mathrm{MC},k=1}F_\sigma(\theta)$ are close to the true gradient $\nabla F(\theta)$, i.e.:* $\left\|\mathbb{E}_{\mathbf{g}\sim\mathcal{N}(0,\mathbf{I}_d)}\left[\widehat{\nabla}^{\mathrm{AT,base}}_{\mathrm{MC},k=1}F_\sigma(\theta)\right] - \nabla F(\theta)\right\| \leq \epsilon$ *and* $\left\|\mathbb{E}_{\mathbf{g}\sim\widehat{P}}\left[\widehat{\nabla}^{\mathrm{AT,asebo}}_{\mathrm{MC},k=1}F_\sigma(\theta)\right] - \nabla F(\theta)\right\| \leq \epsilon$.

### 3.1 Variance reduction via non isotropic sampling

We show now that under sampling strategy given by distribution $\widehat{P}$, the variance of the gradient estimator can be made smaller by choosing the probability parameter $p^t$ appropriately. Denote: $d_{\mathrm{active}} = \dim(\mathcal{L}^{\mathrm{ES}}_{\mathrm{active}})$ and $d_\perp = \dim(\mathcal{L}^{\mathrm{ES},\perp}_{\mathrm{active}})$. Let $\Gamma := \frac{d_{\mathrm{active}}+2}{p^t}s_{\mathbf{U}^{\mathrm{act}}} + \frac{d_\perp+2}{1-p^t}s_{\mathbf{U}^\perp} - \|\nabla F(\theta)\|^2$.

**Theorem 3.2.** *The following holds for $s_{\mathbf{U}^{\mathrm{act}}} = \|(\mathbf{U}^{\mathrm{act}})^\top\nabla F(\theta)\|_2^2$ and $s_{\mathbf{U}^\perp} = \|(\mathbf{U}^\perp)^\top\nabla F(\theta)\|_2^2$:*

*1. The variance of $\widehat{\nabla}^{\mathrm{AT,asebo}}_{\mathrm{MC},k=1}F_\sigma(\theta)$ is close to $\Gamma$, i.e. $|\mathrm{Var}[\widehat{\nabla}^{\mathrm{AT,asebo}}_{\mathrm{MC},k=1}F_\sigma(\theta)] - \Gamma| \leq \epsilon$.*

*2. The choice of $p^t$ that minimizes $\Gamma$ satisfies $p^t_* := \frac{\sqrt{(s_{\mathbf{U}^{\mathrm{act}}})(d_{\mathrm{active}}+2)}}{\sqrt{(s_{\mathbf{U}^{\mathrm{act}}})(d_{\mathrm{active}}+2)}+\sqrt{(s_{\mathbf{U}^\perp})(d_{\mathbf{U}\perp}+2)}}$ and the optimal variance $\mathrm{Var}_{\mathrm{opt}}$ corresponding to $p^t_*$ satisfies: $|\mathrm{Var}_{\mathrm{opt}} - \Delta| \leq \epsilon$ for $\Delta = \left[\sqrt{(s_{\mathbf{U}^{\mathrm{act}}})(d_{\mathrm{active}}+2)} + \sqrt{(s_{\mathbf{U}^\perp})(d_\perp+2)}\right]^2 - \|\nabla F(\theta)\|^2$.*

3. $\mathrm{Var_{opt}} \leq \mathrm{Var}[\widehat{\nabla}_{\mathrm{MC},k=1}^{\mathrm{AT,base}}F_\sigma(\theta)] + \epsilon \underbrace{- \lfloor \sqrt{(s_{\mathbf{U}^\perp})(d_{\mathrm{active}}+2)} - \sqrt{(s_{\mathbf{U}^{\mathrm{act}}})(d_\perp+2)}|^2 - 2\|\nabla F(\theta)\|^2}_{\lambda}$.

*Furthermore, slack variable $\lambda$ is always nonnegative.*

Theorem implies that when $s_{\mathbf{U}^{\mathrm{act}}} = (1-\alpha)\|\nabla F(\theta)\|_2^2$ and $s_{\mathbf{U}^\perp} = \alpha\|\nabla F(\theta)\|_2^2$, for some $\alpha \in (0,1)$, we have: $\mathrm{Var}[\widehat{\nabla}_{\mathrm{MC},k=1}^{\mathrm{AT,base}}F_\sigma(\theta)] \approx (d+1)\|\nabla F(\theta)\|^2$ whereas $\mathrm{Var_{opt}} = \mathcal{O}\left((1-\alpha)(d_{\mathrm{active}}+1)+\alpha(d_\perp+1)\right)$. When $d_{\mathrm{active}} << d$ and $\alpha << 1$: $\mathrm{Var_{opt}} \ll \mathrm{Var}[\widehat{\nabla}_{\mathrm{MC},k=1}^{\mathrm{AT,base}}F_\sigma(\theta)]$.

## 3.2 Adaptive Mirror Descent

In Theorem 3.2 we showed that for appropriate choices of $\mathcal{L}_{\mathrm{active}}^{\mathrm{ES}}$ and $p_t$, the gradient estimator $\widehat{\nabla}_{\mathrm{MC},k=1}^{\mathrm{AT,asebo}}F_\sigma(\theta)$ will have significantly smaller variance than $\widehat{\nabla}_{\mathrm{MC},k=1}^{\mathrm{AT,base}}F_\sigma(\theta)$. In this section we show that Algorithm 2 provides an adaptive way to choose $p^t$. Using tools from online learning theory, we provide regret guarantees that quantify the rate at which this Algorithm 2 minimizes the variance of $\widehat{\nabla}_{\mathrm{MC},k=1}^{\mathrm{AT,asebo}}F_\sigma(\theta)$ and converges to the optimal $p_*^t$.

Let $\mathbf{p}_l^t = \binom{p_l^t}{1-p_l^t}$. The main component $\Gamma$ of the variance of $\widehat{\nabla}_{\mathrm{MC},k=1}^{\mathrm{AT,asebo}}F_\sigma(\theta)$ as a function of $\mathbf{p}_l^t$ equals $\Gamma = \ell(\mathbf{p}_l^t) = \frac{d_{\mathrm{active}}+2}{\mathbf{p}_l^t(1)}s_{\mathbf{U}^{\mathrm{act}}} + \frac{d_\perp+2}{\mathbf{p}_l^t(2)}s_{\mathbf{U}^\perp} - \|\nabla F(\theta)\|^2$ (Theorem 3.2). We have:

**Theorem 3.3.** *Let $\Delta_2$ be the a 2-d simplex. Under assumptions: 1 and 2, if $\sigma < \frac{1}{35}\sqrt{\frac{\epsilon \min(p^t, 1-p^t)}{\tau d^3 \max(L,1)}}$, $\alpha = \frac{2\beta^2}{\sqrt{C[(d_{\mathrm{active}}+2)^2 s_{\mathbf{U}^{\mathrm{act}}}^2 + (d_\perp+2)s_{\mathbf{U}^\perp}^2]}}$ and $\epsilon = \frac{\beta^3}{2C(d+1)}$, Algorithm 2 satisfies:*

$$\frac{1}{C}\mathbb{E}\left[\sum_{l=1}^C \ell(\mathbf{p}_l^t)\right] - \min_{\mathbf{p} \in \beta+(1-2\beta)\Delta_2} \ell(\mathbf{p}) \leq \frac{\mathrm{Var_{opt}}}{\beta^2\sqrt{C}} + \frac{1}{C}$$

# 4 Experiments

In our experiments we use different classes of high-dimensional blackbox functions: RL blackbox functions (where the input is a high-dimensional vector encoding a neural network policy $\pi : \mathcal{S} \to \mathcal{A}$ mapping states $s$ to actions $a$ and the output is the cumulative reward obtained by an agent applying this policy in a particular environment) and functions from the recently open-sourced Nevergrad library [34]. In practice one can setup the hyperparameters used by Algorithm 2 as follows: $\sigma = 0.01, C = 10, \alpha = 0.01, \beta = 0.1, q_0^t = 0.1$. For each algorithm we used $k = 5$ seeds and obtained curves are median-curves with inter-quartile ranges presented as shadowed regions.

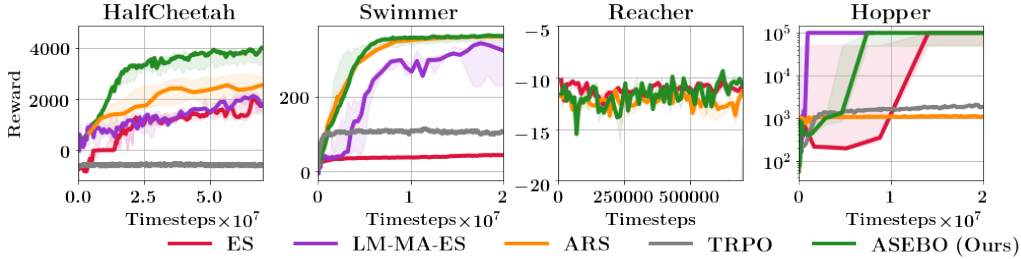

Figure 2: Comparison of different blackbox optimization algorithms on OpenAI Gym tasks. All curves are median-curves obtained from $k = 5$ seeds and with inter-quartile ranges presented as shadowed regions. For Reacher we present only 3 curves since LM-MA-ES and TRPO did not learn.

## 4.1 RL blackbox functions

We used the following environments from the OpenAI Gym library: Swimmer-v2, HalfCheetah-v2, Walker2d-v2, Reacher-v2, Pusher-v2 and Thrower-v2. In all experiments we used policies

encoded by neural network architectures of two hidden layers and with $\mathrm{tanh}$ nonlinearities, with $> 100$ parameters. For gradient-based optimization we use $\mathrm{Adam}$. For this class of blackbox functions we compared ASEBO with other generic blackbox methods as well as those specializing in optimizing RL blackbox functions $F$, namely: **(1)** CMA-ES variants; we compare against two recently introduced algorithms designed for high-dimensional settings (we use the implementation of VkD-CMA-ES in the $\mathrm{pycma}$ open-source implementation from $\mathrm{https://github.com/CMA\text{-}ES/pycma}$), and that of LM-MA-ES from [26]), **(2)** Augmented Random Search (ARS) [25] (we use implementation released by the authors at $\mathrm{http://github.com/modestyachts/ARS}$), **(3)** Proximal Policy Optimization (PPO) [32] and Trust Region Policy Optimization (TRPO) [31] (we use $\mathrm{OpenAI}$ baseline implementation [12]). The results for four environments are on Fig. 2.

Table 1: Median rewards obtained across $k = 5$ seeds for seven RL environments. For each environment the top two performing algorithms are bolded, while the bottom two are shown in red.

| Environment | Timesteps | Median reward after # timesteps | | | | | | |
|---|---|---|---|---|---|---|---|---|
| | | ASEBO | ES | ARS | VkD-CMA | LM-MA | TRPO | PPO |
| HalfCheetah | $5.10^7$ | **3821** | 1530 | **2420** | -144 | 1632 | -512 | 1514 |
| Swimmer | $10^7$ | **358** | 36 | 348 | **367** | 297 | 110 | 52 |
| Walker2d | $5.10^7$ | 9941 | 347 | 1112 | 1 | **18065** | 3011 | 2377 |
| Hopper | $10^7$ | 99949 | 626 | 1091 | 42 | **100199** | 1663 | 1310 |
| Reacher | $10^5$ | $-11$ | $-10$ | -12 | -1391 | -173 | -112 | -196 |
| Pusher | $10^5$ | $-46$ | -48 | $-45$ | -1001 | -467 | -120 | -316 |
| Thrower | $10^5$ | $-89$ | -96 | -90 | -796 | -737 | $-85$ | -175 |

Sampling complexity is measured in the number of timesteps (environment transitions) used by the algorithms. ASEBO is the only algorithm that performs consistently across all seven environments (see: Table 1), outperforming CMA-ES variants on all tasks aside from VkD-CMA-ES on Swimmer and LM-MA-ES on Walker2d. For environments such as Reacher, Thrower and Pusher, these methods perform poorly, drastically underperforming even Vanilla ES. On Fig. 3, we demonstrate the common problem of state-of-the-art CMA-ES methods: if the number of samples $n$ is not carefully tuned, the algorithm does not learn. ASEBO does not have this problem since $n$ is learned on-the-fly.

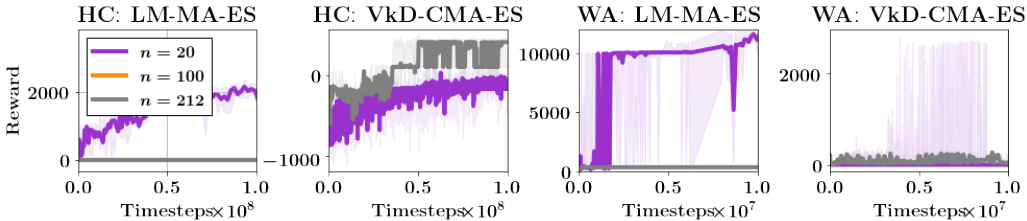

Figure 3: Sensitivity analysis for CMA-ES variants on the $\mathrm{HalfCheetah}$ (HC) and Walker2d (WA) tasks. In each setting, we run $k = 5$ seeds, solely changing the number of samples per iteration (or *population size*) $n$.

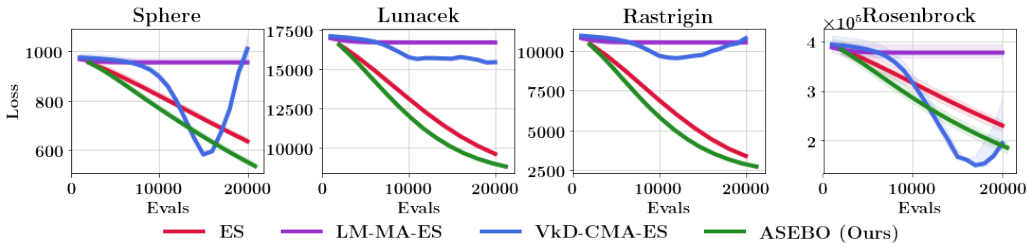

Figure 4: Comparison of median-curves obtained from $k = 5$ seeds for different algorithms on $\mathrm{Nevergrad}$ functions [34]. Inter-quartile ranges are presented as shadowed regions.

### 4.2 Nevergrad **blackbox functions**

We tested functions: sphere, rastrigin, rosenbrock and lunacek (from the class of Bi-Rastrigin/Lunacek's No.02 functions). All tested functions are 1000-dimensional. The results are presented on Fig. 4. ASEBO is the most reliable method across different functions.

## 5 Conclusion

We proposed a new algorithm for optimizing high-dimensional blackbox functions. ASEBO adjusts on-the-fly the strategy of choosing gradient sensing directions to the hardness of the problem at the current stage of optimization and can be applied for both RL and non-RL problems. We provided theoretical guarantees for our method and exhaustive empirical validation.

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
