[Supplementary Material]

# APPENDIX: From Complexity to Simplicity: Adaptive ES-Active Subspaces for Blackbox Optimization

## 6 Theoretical Results

Throughout this section we will assume the sensings directions $\{\mathbf{g}_i\}$ at time $t$ are sampled from one of the following families of distributions:

$$\widehat{P} = \begin{cases} \mathbf{g} \sim \mathcal{N}(0, \mathbf{I}_{\mathcal{L}_{\text{active}}^{\text{ES}}}) & \text{with probability } p^t \\ \mathbf{g} \sim \mathcal{N}(0, \mathbf{I}_{\mathcal{L}_{\text{active}}^{\text{ES},\perp}}) & \text{with probability } 1 - p^t \end{cases}$$

Where $p^t$ is a probability parameter with values in $[0, 1]$.

Denote an by $\mathbf{U}^{\text{act}} \in \mathbb{R}^{d \times d_{\text{active}}}$ an orthonormal basis of the active subspace $\mathcal{L}_{\text{active}}^{\text{ES}}$ and $\mathbf{U}^{\perp} \in \mathbb{R}^{d \times (d - d_{\text{active}})}$ an orthonormal basis of $\mathcal{L}_{\text{active}}^{\text{ES},\perp}$.

Let's start by computing the covariance matrix of $\widehat{P}$:

$$\mathbb{E}_{\mathbf{g} \sim P_i}\left[\mathbf{g}\mathbf{g}^{\top}\right] = \underbrace{\left(p^t \mathbf{U}^{\text{act}}(\mathbf{U}^{\text{act}})^{\top} + (1 - p^t)\mathbf{U}^{\perp}(\mathbf{U}^{\perp})^{\top}\right)}_{\mathbf{C}_1}$$

In order to simplify the notation of the proofs in this section we use the following conventions:

$$z_{ES} = \widehat{\nabla}_{\text{MC},k=1}^{\text{AT,base}} F_{\sigma}(\theta)$$
$$z_1 = \widehat{\nabla}_{\text{MC},k=1}^{\text{AT,asebo}} F_{\sigma}(\theta)$$

Where $z_1$ is the ASEBO gradient estimator resulting form using sampling mechanism $\widehat{P}$.

**Notational simplification** To simplify notation we also use $\mathbf{U}$ instead of $\mathbf{U}^{\text{act}}$, $\mathbf{I}_{\mathbf{U}}$ instead of $\mathbf{I}_{\mathcal{L}_{\text{active}}^{\text{ES}}}$ and $\mathbf{I}_{\mathbf{U}^{\perp}}$ instead of $\mathbf{I}_{\mathcal{L}_{\text{active}}^{\text{ES},\perp}}$

Let $\epsilon > 0$ be the precision parameter. We choose $\sigma$ with the goal of making the bias between the expectation of our gradient estimators and the true gradient of $F$ smaller than $\epsilon$. Throughout this section we assume $\sigma$ is small enough:

$$0 < \sigma < \frac{1}{35}\sqrt{\frac{\epsilon \min(p^t, 1 - p^t)}{\tau d^3 \max(L, 1)}}$$

### 6.1 Gradient Estimators, their bias and their variance.

In this section we aim to produce theoretical guarantees regarding the bias and variance of our proposed gradient estimators. We show that under the right assumptions, the isotropic and non isotropic versions of the evolution Strategies estimators have small bias, and

We make the following assumptions on $F$:

> **Assumption 1**. $F$ is $L-$Lipschitz. For all $\theta, \theta' \in \mathbb{R}^d$, $|F(\theta) - F(\theta')| \leq L\|\theta - \theta'\|$.
>
> **Assumption 2**. $F$ has a $\tau$-smooth third order derivative tensor, so that $F(\theta + \sigma\mathbf{g}) = F(\theta) + \sigma\nabla F(\theta)^{\top}\mathbf{g} + \frac{\sigma^2}{2}\mathbf{g}^{\top}H(\theta)\mathbf{g} + \frac{1}{6}\sigma^3 F'''(\theta)[v, v, v]$ with $v \in [0, \mathbf{g}]$ satisfying $|F'''(\theta)[v, v, v] \leq \tau\|v\|^3 \leq \tau\|\mathbf{g}\|^3$.

Let $d_{\text{active}}$ and $d_{\perp}$ denote the dimensionality of $\mathcal{L}_{(active)}$ and $\mathcal{L}_{\perp}$ respectively.

Under these assumptions, $\frac{F(\theta_t + \sigma\mathbf{g}) - F(\theta_t - \sigma\mathbf{g})}{2\sigma} = \left(\mathbf{g}^{\top}\nabla F(\theta_t)\right) + \xi_{\mathbf{g}}(\theta_t)$ such that $\xi_{\mathbf{g}}(\theta_t) \leq \frac{\tau}{6}\sigma^2\|\mathbf{g}\|^3$, uniformly over all $\theta_t$. We relax the constants slightly. If $F$'s third order derivative tensor is smooth with constant $\tau$:

$$\left| \frac{F(\theta_t + \sigma \mathbf{g}) - F(\theta_t - \sigma \mathbf{g})}{2\sigma} - \mathbf{g}^\top \nabla F(\theta_t) \right| \leq \tau \sigma^2 \|\mathbf{g}\|^3.$$

Recall the following definitions:

- **Evolution Strategies Gradient**. Let $\mathbf{g} \sim \mathcal{N}(0, \mathbf{I})$. The ES gradient is defined as $z_{ES} = \frac{F(\theta_t + \sigma \mathbf{g}) - F(\theta_t - \sigma \mathbf{g})}{2\sigma} \mathbf{g}$.

- $\widehat{P}$ **Nonisotropic Gradient.**. Let $\mathbf{g} \sim \widehat{P}$. The $\widehat{P}$ gradient is defined as $z_1 = \mathbf{C}_1^{-1} \frac{F(\theta_t + \sigma \mathbf{g}) - F(\theta_t - \sigma \mathbf{g})}{2\sigma} \mathbf{g}$.

The following inequalitites hold:

$$\|\xi_{\mathbf{g}}(\theta_t)\mathbf{g}\|^2 \leq \frac{\tau}{6} \sigma^2 \|\mathbf{g}\|^4$$

$$\|\mathbb{E}_{\mathbf{g} \sim \mathcal{N}(0,\mathbf{I})} [\xi_{\mathbf{g}}(\theta_t)\mathbf{g}] \|^2 \leq \frac{\sigma^4 \tau^2}{36} \left( \mathbb{E}_{\mathbf{g} \sim \mathcal{N}(0,\mathbf{I})} [\|\mathbf{g}\|^4] \right)^2 \leq \frac{\sigma^4 \tau^2 d^4}{4}$$

$$\|\mathbb{E}_{\mathbf{g} \sim \mathcal{N}(0,\mathbf{I_U})} [\xi_{\mathbf{g}}(\theta_t)\mathbf{g}] \|^2 \leq \frac{\sigma^4 \tau^2}{36} \left( \mathbb{E}_{\mathbf{g} \sim \mathcal{N}(0,\mathbf{I_{U^\perp}})} [\|\mathbf{g}\|^4] \right)^2 \leq \frac{\sigma^4 \tau^2 d^4_{\text{active}}}{4}$$

$$\|\mathbb{E}_{\mathbf{g} \sim \mathcal{N}(0,\mathbf{I_U})} [\xi_{\mathbf{g}}(\theta_t)\mathbf{g}] \|^2 \leq \frac{\sigma^4 \tau^2}{36} \left( \mathbb{E}_{\mathbf{g} \sim \mathcal{N}(0,\mathbf{I_{U^\perp}})} [\|\mathbf{g}\|^4] \right)^2 \leq \frac{\sigma^4 \tau^2 d^4_\perp}{4}$$

**Bounding the Bias** The first result in this section is to show that under the right conditions the ES gradient estimators in both the isotropic and non isotropic cases can be close to the true gradient provided the function satisfies Assumptions 1 and 2. Theorem 6.1 deals with the isotropic case and Theorem 6.2 with the non isotropic case. The combination of these results yields the proof of Lemma 3.1 in the main text.

**Theorem 6.1.** *The evolution strategies gradient estimator $z_{ES}$ satisfies:*

$$\|\mathbb{E}_{\mathbf{g} \sim \mathcal{N}(0,\mathbf{I})} [z_{ES}] - \nabla F(\theta_t)\| \leq 3\tau \sigma^2 d^2 \tag{4}$$

*If $\sigma < \frac{1}{35} \sqrt{\frac{\epsilon \min(p^t, 1-p^t)}{\tau d^3 \max(L, 1)}}$:*

$$\|\mathbb{E}_{\mathbf{g} \sim \mathcal{N}(0,\mathbf{I})} [z_{ES}] - \nabla F(\theta_t)\| \leq \epsilon \tag{5}$$

*Proof.* Notice that $\|\mathbf{g}\|^4 = (\sum_{i=1}^d \mathbf{g}(i)^2)^2 \leq d \sum_{i=1}^d \mathbf{g}(i)^4$. Where we denote $\mathbf{g}(i)$ as the $i$−th entry of the $d$−dimensional vector $\mathbf{g} \in \mathbb{R}^d$. Since $\mathbb{E}[\mathbf{g}(i)^4] = 3$ for all $i$:

$$\mathbb{E}_{\mathbf{g} \sim \mathcal{N}(0,\mathbf{I})} [\|\mathbf{g}\|^4] \leq 3d^2$$

And therefore:

$$\left\| \mathbb{E}_{\mathbf{g} \sim \mathcal{N}(0,\mathbf{I})} \left[ \frac{F(\theta_t + \sigma \mathbf{g}) - F(\theta_t - \sigma \mathbf{g})}{2\sigma} \mathbf{g} \right] - \nabla F(\theta_t) \right\| \leq \tau \sigma^2 \mathbb{E}_{\mathbf{g} \sim \mathcal{N}(0,\mathbf{I})} [\|\mathbf{g}\|^4] \leq 3\tau \sigma^2 d^2$$

$\square$

A similar result holds for the $z_1$ gradient.

**Theorem 6.2.** *The non isotropic $\widehat{P}$ gradient estimator satisfies:*

$$\|\mathbb{E}_{\mathbf{g} \sim \widehat{P}} [z_1] - \nabla F(\theta_t)\| \leq \frac{3\sigma^2 \tau}{p^t} d^2_{\text{active}} + \frac{3\sigma^2 \tau}{1 - p^t} d^2_\perp$$

*If $\sigma < \frac{1}{35} \sqrt{\frac{\epsilon \min(p^t, 1-p^t)}{\tau d^3 \max(L, 1)}}$:*

$$\|\mathbb{E}_{\mathbf{g} \sim \widehat{P}} [z_1] - \nabla F(\theta_t)\| \leq \epsilon$$

*Proof.* Expanding $\mathbb{E}_{\mathbf{g}\sim\widehat{P}}[z_1]$ yields:

$$
\begin{aligned}
\mathbb{E}_{\mathbf{g}\sim\widehat{P}}[z_1] &= \mathbf{C}_1^{-1}\mathbb{E}_{\mathbf{g}\sim\widehat{P}}\left[\frac{F(\theta_t+\sigma\mathbf{g})-F(\theta_t-\sigma\mathbf{g})}{2\sigma}\mathbf{g}\right] \\
&= \mathbf{C}_1^{-1}\mathbb{E}_{\mathbf{g}\sim\widehat{P}}\left[\mathbf{g}\mathbf{g}^\top\nabla F(\theta_t)+\xi_{\mathbf{g}}(\theta_t)\mathbf{g}\right] \\
&= \nabla F(\theta_t) + \frac{1}{p^t}\mathbb{E}_{\mathbf{g}\sim\mathcal{N}(0,\mathbf{I}_\mathbf{U})}[\xi_{\mathbf{g}}(\theta_t)\mathbf{g}] + \frac{1}{1-p^t}\mathbb{E}_{\mathbf{g}\sim\mathcal{N}(0,\mathbf{I}_{\mathbf{U}\perp})}[\xi_{\mathbf{g}}(\theta_t)\mathbf{g}]
\end{aligned}
$$

By a similar argument as in the proof of Theorem 6.1:

$$
\|\mathbb{E}_{\mathbf{g}\sim\mathcal{N}(0,\mathbf{I}_\mathbf{U})}[\xi_{\mathbf{g}}(\theta_t)\mathbf{g}]\| \le 3\tau\sigma^2 d_{\text{active}}^2
$$
$$
\|\mathbb{E}_{\mathbf{g}\sim\mathcal{N}(0,\mathbf{I}_{\mathbf{U}\perp})}[\xi_{\mathbf{g}}(\theta_t)\mathbf{g}]\| \le 3\tau\sigma^2 d_\perp^2
$$

The result follows. $\qquad\square$

**Towards bounding the variance**  We start by showing how under the right assumptions the expected squared norm of the ES gradients are also bounded away from the squared norms of the true gradients. The distance between the square norms of the expectation of the ES gradient and the true gradient of $F$ are also bounded. Theorem 6.3 deals with the isotropic ES estimator and Theorem 6.4 with its non isotropic counterpart:

**Theorem 6.3.** *If $F$ satisfies Assumption 1 and 2:*

$$
\left|\left\|\mathbb{E}_{\mathbf{g}\sim\mathcal{N}(0,\mathbf{I})}[z_{ES}]\right\|^2 - \|\nabla F(\theta_t)\|^2\right| \le 105\tau^2\sigma^4 d^4 + 6\tau\sigma^2 Ld^2 \tag{6}
$$

*If $\sigma < \frac{1}{35}\sqrt{\frac{\epsilon\min(p^t,1-p^t)}{\tau d^3\max(L,1)}}$:*

$$
\left|\left\|\mathbb{E}_{\mathbf{g}\sim\mathcal{N}(0,\mathbf{I})}[z_{ES}]\right\|^2 - \|\nabla F(\theta_t)\|^2\right| \le \epsilon \tag{7}
$$

*Proof.*

$$
\begin{aligned}
\left|\left\|\mathbb{E}_{\mathbf{g}\sim\mathcal{N}(0,\mathbf{I})}\left[\frac{F(\theta_t+\sigma\mathbf{g})-F(\theta_t-\sigma\mathbf{g})}{2\sigma}\mathbf{g}\right]\right\|^2 - \|\nabla F(\theta_t)\|^2\right| &\le \tau^2\left(\sigma^2\mathbb{E}_{\mathbf{g}\sim\mathcal{N}(0,\mathbf{I})}\left[\|\mathbf{g}\|^4\right]\right)^2 + \\
&\quad 2\tau\sigma^2\|\nabla F(\theta_t)\|\|\mathbb{E}_{\mathbf{g}\sim\mathcal{N}(0,\mathbf{I})}\left[\|\mathbf{g}\|^4\right] \\
&\le 105\tau^2\sigma^4 d^4 + 6\tau\sigma^2 Ld^2
\end{aligned}
$$

$\qquad\square$

**Theorem 6.4.** *If $F$ satisfies Assumption 1 and 2:*

$$
\begin{aligned}
\left|\left\|\mathbb{E}_{\mathbf{g}\sim\widehat{P}}[z_1]\right\|^2 - \|\nabla F(\theta_t)\|^2\right| &\le \frac{1}{(p^t)^2}\frac{\sigma^4\tau^2 d_{\text{active}}^4}{4} + \frac{1}{(1-p^t)^2}\frac{\sigma^4\tau^2 d_\perp^4}{4} + \frac{2}{p^t}L\frac{\sigma^2\tau d_{\text{active}}^2}{4} + \\
&\quad \frac{2}{1-p^t}L\frac{\sigma^2\tau d_\perp^2}{4} + \frac{2}{p^t(1-p^t)}\frac{\sigma^4\tau^2 d_{\text{active}}^2 d_\perp^2}{16}
\end{aligned}
$$

*If $\sigma < \frac{1}{35}\sqrt{\frac{\epsilon\min(p^t,1-p^t)}{\tau d^3\max(L,1)}}$:*

$$
\left|\left\|\mathbb{E}_{\mathbf{g}\sim\widehat{P}}[z_1]\right\|^2 - \|\nabla F(\theta_t)\|^2\right| \le \epsilon
$$

*Proof.* Consider the following expansion of $\mathbb{E}[z_1]$.

$$\|\mathbb{E}_{\mathbf{g}\sim\widehat{P}}[z_1]\|^2 = \|\nabla F(\theta_t)\|^2 + \frac{1}{(p^t)^2}\|\mathbb{E}_{\mathbf{g}\sim\mathcal{N}(0,\mathbf{I_U})}[\xi_{\mathbf{g}}(\theta_t)\mathbf{g}]\|^2 + \left(\frac{1}{1-p^t}\right)^2\|\mathbb{E}_{\mathbf{g}\sim\mathcal{N}(0,\mathbf{I_{U^\perp}})}[\xi_{\mathbf{g}}(\theta_t)\mathbf{g}] +$$

$$\frac{2}{p^t}\langle\nabla F(\theta_t),\mathbb{E}_{\mathbf{g}\sim\mathcal{N}(0,\mathbf{I_U})}[\xi_{\mathbf{g}}(\theta_t)\mathbf{g}]\rangle + \frac{2}{1-p^t}\langle\nabla F(\theta_t),\mathbb{E}_{\mathbf{g}\sim\mathcal{N}(0,\mathbf{I_{U^\perp}})}[\xi_{\mathbf{g}}(\theta_t)\mathbf{g}]\rangle +$$

$$\frac{2}{p^t(1-p^t)}\langle\nabla\mathbb{E}_{\mathbf{g}\sim\mathcal{N}(0,\mathbf{I_U})}[\xi_{\mathbf{g}}(\theta_t)\mathbf{g}],\mathbb{E}_{\mathbf{g}\sim\mathcal{N}(0,\mathbf{I_{U^\perp}})}[\xi_{\mathbf{g}}(\theta_t)\mathbf{g}]\rangle$$

And therefore by Cauchy Schwartz:

$$\left|\|\mathbb{E}_{\mathbf{g}\sim\widehat{P}}[z_1]\|^2 - \|\nabla F(\theta_t)\|^2\right| \leq \frac{1}{(p^t)^2}\frac{\sigma^4\tau^2 d_{\text{active}}^4}{4} + \frac{1}{(1-p^t)^2}\frac{\sigma^4\tau^2 d_\perp^4}{4} +$$

$$\frac{2}{p^t}L\frac{\sigma^2\tau d_{\text{active}}^2}{4} + \frac{2}{1-p^t}L\frac{\sigma^2\tau d_\perp^2}{4} + \frac{2}{p^t(1-p^t)}\frac{\sigma^4\tau^2 d_{\text{active}}^2 d_\perp^2}{16}$$

As desired. $\qquad\qquad\square$

**Bounding the variance of $z_{ES}$ and $z_1$.** We have now the necessary ingredients for bounding the variance of the ES isotropic and non isotropic estimators. We start by showing in theorem 6.5 that the variance of the isotropic estimator is roughly of the order of $(d+1)\|\nabla F(\theta_t)\|^2$. In contrast, Theorem 6.6 characterizes the variance of $z_1$ the non isotropic ES gradient estimator in terms of the $\nabla F(\theta_t)$ decomposition along the subspaces spanned by $\mathbf{U}$ and $\mathbf{U}^\perp$. In the following section 6.2 we show that with an appropriate choice of the probabilities $p^t$, $1-p^t$, and provided the subspace decomposition is adequate, the variance of the non isotropic gradient estimator can be much smaller than the variance of the $z_{ES}$.

**Theorem 6.5.** *If F satisfies Assumption 1 and 2, the variance of the ES estimator satisfies:*

$$|Var_{ES} - (d+1)\|\nabla F(\theta_t)\|^2| \leq 105\tau^2\sigma^4 d^4 + 6\tau\sigma^2 L d^2 + 15 d^3\sigma^2 L\tau + 105\tau^2\sigma^4 d^4$$

*If $\sigma < \frac{1}{35}\sqrt{\frac{\epsilon\min(p^t,1-p^t)}{\tau d^3\max(L,1)}}$:*

$$|Var_{ES} - (d+1)\|\nabla F(\theta_t)\|^2| \leq \epsilon$$

*Proof.* The second moment of the ES estimator satisfies:

$$\mathbb{E}_{\mathbf{g}\sim\mathcal{N}(0,\mathbf{I})}\left[z_{ES}^\top z_{ES}\right] = \mathbb{E}_{\mathbf{g}\sim\mathcal{N}(0,\mathbf{I})}\left[\frac{(F(\theta_t+\sigma\mathbf{g})-F(\theta_t-\sigma\mathbf{g}))^2)}{2^2\sigma^2}\mathbf{g}^\top\mathbf{g}\right]$$

$$= \mathbb{E}_{\mathbf{g}\sim\mathcal{N}(0,\mathbf{I})}\left[\left(\mathbf{g}^\top\nabla F(\theta_t)+\xi_{\mathbf{g}}(\theta_t)\right)^2\mathbf{g}^\top\mathbf{g}\right]$$

$$= \mathbb{E}_{\mathbf{g}\sim\mathcal{N}(0,\mathbf{I})}\left[\nabla F(x_t)^\top\mathbf{gg}^\top\mathbf{gg}^\top\nabla F(\theta_t)+2\nabla F(\theta_t)^\top\mathbf{gg}^\top\mathbf{g}\xi_{\mathbf{g}}(\theta_t)+\xi_{\mathbf{g}}(\theta_t)^2\mathbf{g}^\top\mathbf{g}\right]$$

$$= (d+2)\|\nabla F(\theta_t)\|^2 + 2\mathbb{E}_{\mathbf{g}\sim\mathcal{N}(0,\mathbf{I})}\left[\nabla F(\theta_t)^\top\mathbf{gg}^\top\mathbf{g}\xi_{\mathbf{g}}(\theta_t)\right] +$$

$$\mathbb{E}_{\mathbf{g}\sim\mathcal{N}(0,\mathbf{I})}\left[\xi_{\mathbf{g}}(\theta_t)^2\mathbf{g}^\top\mathbf{g}\right]$$

Under Assumption 1 and 2, the following bound for the second and third terms of the last equality holds:

$$\left|\mathbb{E}_{\mathbf{g}\sim\mathcal{N}(0,\mathbf{I})}\left[\nabla F(\theta_t)^\top\mathbf{gg}^\top\mathbf{g}\xi_{\mathbf{g}}(\theta_t)\right]\right| \leq \mathbb{E}_{\mathbf{g}\sim\mathcal{N}(0,\mathbf{I})}\left[\|\nabla F(\theta_t)\|\|\mathbf{g}\|^6\right] \leq 15 d^3\sigma^2 L\tau$$

And:

$$\left|\mathbb{E}_{\mathbf{g}\sim\mathcal{N}(0,\mathbf{I})}\left[\xi_{\mathbf{g}}(\theta_t)^2\mathbf{g}^\top\mathbf{g}\right]\right| \leq \tau^2\sigma^4\mathbb{E}_{\mathbf{g}\sim\mathcal{N}(0,\mathbf{I})}\left[\|\mathbf{g}\|^8\right] \leq 105\tau^2\sigma^4 d^4$$

Therefore:

$$\text{Var}_{\text{ES}} = \underbrace{\mathbb{E}_{\mathbf{g}\sim\mathcal{N}(0,\mathbf{I})}\left[\frac{(F(\theta_t+\sigma\mathbf{g})-F(\theta_t-\sigma\mathbf{g}))^2)}{2^2\sigma^2}\mathbf{g}^\top\mathbf{g}\right]}_{\diamondsuit} - \underbrace{\left\|\mathbb{E}_{\mathbf{g}\sim\mathcal{N}(0,\mathbf{I})}\left[\frac{F(\theta_t+\sigma\mathbf{g})-F(\theta_t-\sigma\mathbf{g})}{2\sigma}\mathbf{g}\right]\right\|^2}_{\spadesuit}$$

After coalescing the bounds dervied in the preceeding section, we can obtain the following bound on the term $\diamondsuit$:

$$\left|\diamondsuit - (d+2)\|\nabla F(\theta_t)\|^2\right| \le 15d^3\sigma^2 L\tau + 105\tau^2\sigma^4 d^4$$

Notice that by virtue of 6, the following bound on term $\spadesuit$ of the previous equation holds:

$$\left|\spadesuit - \|\nabla F(\theta_t)\|^2\right| \le 105\tau^2\sigma^4 d^4 + 6\tau\sigma^2 L d^2$$

Combining these two inequalities the result follows.

$\square$

A similar theorem holds for $z_1$.

**Theorem 6.6.** *Let* $\Gamma = \left(\frac{d_{\text{active}}+2}{p^t}\|\mathbf{U}^\top\nabla F(\theta_t)\|^2 + \frac{d_\perp+2}{1-p^t}\|(\mathbf{U}^\perp)^\top F(\theta_t)\|^2 - \|\nabla F(\theta_t)\|^2\right).$

$$\left|Var_{\widehat{P}} - \Gamma\right| \le \frac{1}{p^t}\left(15d_{\text{active}}^3\sigma^2 L\tau + 105\tau^2\sigma^4 d_{\text{active}}^4\right) +$$
$$\frac{1}{1-p^t}\left(15d_\perp^3\sigma^2 L\tau + 105\tau^2\sigma^4 d_\perp^4\right) +$$
$$\frac{1}{(p^t)^2}\frac{\sigma^4\tau^2 d_{\text{active}}^4}{4} + \frac{1}{(1-p^t)^2}\frac{\sigma^4\tau^2 d_\perp^4}{4} +$$
$$\frac{2}{p^t}L\frac{\sigma^2\tau d_{\text{active}}^2}{4} + \frac{2}{1-p^t}L\frac{\sigma^2\tau d_\perp^2}{4} +$$
$$\frac{2}{p^t(1-p^t)}\frac{\sigma^4\tau^2 d_{\text{active}}^2 d_\perp^2}{16}$$

*If* $\sigma < \frac{1}{35}\sqrt{\frac{\epsilon\min(p^t,1-p^t)}{\tau d^3\max(L,1)}}$:

$$\left|Var_{\widehat{P}} - \left(\frac{d_{\text{active}}+2}{p^t}\|\mathbf{U}^\top\nabla F(\theta_t)\|^2 + \frac{d_\perp+2}{1-p^t}\|(\mathbf{U}^\perp)^\top F(\theta_t)\|^2 - \|\nabla F(\theta_t)\|^2\right)\right| \le \epsilon$$

*Proof.* The second moment of $z_1$ satisfies:

$$\mathbb{E}_{\mathbf{g}\sim\widehat{P}}\left[z_1^\top z_1\right] = \frac{1}{p^t}\mathbb{E}_{\mathbf{g}\sim\mathcal{N}(0,\mathbf{I}_\mathbf{U})}\left[\frac{(F(\theta_t+\sigma\mathbf{g})-F(\theta_t-\sigma\mathbf{g}))^2)}{2^2\sigma^2}\mathbf{g}^\top\mathbf{g}\right] +$$
$$\frac{1}{1-p^t}\mathbb{E}_{\mathbf{g}\sim\mathcal{N}(0,\mathbf{I}_{\mathbf{U}^\perp})}\left[\frac{(F(\theta_t+\sigma\mathbf{g})-F(\theta_t-\sigma\mathbf{g}))^2)}{2^2\sigma^2}\mathbf{g}^\top\mathbf{g}\right]$$

Notice that:

$$\text{Var}_{\widehat{P}} = \underbrace{\mathbb{E}_{\mathbf{g}\sim\widehat{P}}\left[\frac{(F(\theta_t+\sigma\mathbf{g})-F(\theta_t-\sigma\mathbf{g}))^2)}{2^2\sigma^2}\mathbf{g}^\top\mathbf{C}_1^{-2}\mathbf{g}\right]}_{\diamondsuit} - \underbrace{\left\|\mathbb{E}_{\mathbf{g}\sim\widehat{P}}\left[\frac{F(\theta_t+\sigma\mathbf{g})-F(\theta_t-\sigma\mathbf{g})}{2\sigma}\mathbf{C}_1^{-1}\mathbf{g}\right]\right\|^2}_{\spadesuit}$$

By a similar argument as that in the previous theorem, we conclude:

$$\left| \diamondsuit - \frac{d_{\text{active}} + 2}{p^t} \|\mathbf{U}^\top \nabla F(\theta_t)\|^2 - \frac{(d_{V_\perp} + 2)}{1 - p^t} \|(\mathbf{U}^\perp)^\top \nabla F(\theta_t)\|^2 \right| \leq \frac{1}{p^t} \left( 15 d_{\text{active}}^3 \sigma^2 L\tau + 105\tau^2 \sigma^4 d_{\text{active}}^4 \right) +$$
$$\frac{1}{1 - p^t} \left( 15 d_\perp^3 \sigma^2 L\tau + 105\tau^2 \sigma^4 d_\perp^4 \right)$$

By Theorem 6.4:

$$\left| \spadesuit - \|\nabla F(\theta_t)\|^2 \right| \leq \frac{1}{(p^t)^2} \frac{\sigma^4 \tau^2 d_{\text{active}}^4}{4} + \frac{1}{(1 - p^t)^2} \frac{\sigma^4 \tau^2 d_\perp^4}{4} + \frac{2}{p^t} L \frac{\sigma^2 \tau d_{\text{active}}^2}{4} + \frac{2}{1 - p^t} L \frac{\sigma^2 \tau d_\perp^2}{4} +$$
$$\frac{2}{p^t(1 - p^t)} \frac{\sigma^4 \tau^2 d_{\text{active}}^2 d_\perp^2}{16}$$

$\square$

## 6.2 Variance reduction via non isotropic sampling

The first result of this section is to condense the theorems in the previous sections into a single result (see Theorem 6.7). Lemma 6.8 then shows what the variance corresponding to the optimal choice of parameter $p^t$ is. Theorem 6.9 then provides conditions under which the approximate variance (without considering the bias terms) corresponding to the optimal non isotropic estimator is smaller than the variance of the isotropic one. Finally Thoerem 6.10 takes into account the bias and states the final reuslt of this section. The combination of these results yield the proof of Theorem 3.2 in the main section of the paper.

**Theorem 6.7.** *Let* $\epsilon > 0$. *If* $\sigma < \frac{1}{35} \sqrt{\frac{\epsilon \min(p^t, 1 - p^t)}{\tau d^3 \max(L, 1)}}$ *then:*

$$\left\| \mathbb{E}_{\mathbf{g} \sim \mathcal{N}(0, \mathbf{I})} \left[ z_{ES} \right] - \nabla F(\theta_t) \right\| \leq \epsilon \tag{8}$$
$$\left\| \mathbb{E}_{\mathbf{g} \sim \widehat{P}} \left[ z_1 \right] - \nabla F(\theta_t) \right\| \leq \epsilon \tag{9}$$

*and*

$$\left| Var_{ES} - (d + 1) \|\nabla F(\theta_t)\|^2 \right| \leq \epsilon \tag{10}$$
$$\left| Var_{\widehat{P}} - \left( \frac{d_{\text{active}} + 2}{p^t} \|\mathbf{U}^\top \nabla F(\theta_t)\|^2 + \frac{d_\perp + 2}{1 - p^t} \|(\mathbf{U}^\perp)^\top F(\theta_t)\|^2 - \|\nabla F(\theta_t)\|^2 \right) \right| \leq \epsilon \tag{11}$$

We say that in this case:

$$\text{Var}_{\widehat{P}} \approx \underbrace{\left( \frac{d_{\text{active}} + 2}{p^t} \|\mathbf{U}^\top \nabla F(\theta_t)\|^2 + \frac{d_\perp + 2}{1 - p^t} \|(\mathbf{U}^\perp)^\top F(\theta_t)\|^2 - \|\nabla F(\theta_t)\|^2 \right)}_{\text{Var}_{\widehat{P}}^M}$$

and $\text{Var}_{ES} \approx (d + 1)\|\nabla F(\theta_t)\|^2$. We refer to $\text{Var}_{\widehat{P}}^M$ as the "main component" of the variance $\text{Var}_{\widehat{P}}$. Similarly we define $\text{Var}_{ES}^M = (d + 1)\|\nabla F(\theta_t)\|^2$ and use the same name, "main component" of the variance $\text{Var}_{ES}^M$.

The optimal $p^t$, that which minimizes $\text{Var}_{\widehat{P}}^M$ equals:

$$(p^t)^* = \frac{\| (\nabla F(\theta_t))_{active} \| \sqrt{d_{\text{active}} + 2}}{\| (\nabla F(\theta_t))_{active} \| \sqrt{d_{\text{active}} + 2} + \| (\nabla F(\theta))_\perp \| \sqrt{d_\perp + 2}}$$

*Proof.* Roughly the same argument as above yields the desired result. □

**Lemma 6.8.** *The optimal variance* $Var_{\widehat{P}^*}^M$ *corresponding to* $(p^t)^*$ *equals:*

$$\left[\|\,(\nabla F(\theta_t))_{active}\,\|\sqrt{d_{\text{active}}+2}+\|\,(\nabla F(\theta_t))_{\perp}\,\|\sqrt{d_{\perp}+2}\right]^2-\|\nabla F(\theta_t)\|^2 \qquad (12)$$

*Proof.* The statement follows directly from substituting the expression for $(p^t)^*$ into the variance formula. □

**Theorem 6.9.** $Var_{\widehat{P}^*}^M \leq Var_{ES}^M$ *if*

$$|\sqrt{d_{\text{active}}+2}\|\,(\nabla F(\theta_t))_{\perp}\,\|-\sqrt{d_{\perp}+2}\|\nabla F(\theta_t)_{active}\||\geq \sqrt{2}\|\nabla F(\theta_t)\|$$

*Proof.* By definition, $\text{Var}_{\widehat{P}^*}^M < \text{Var}_{ES}^M$ if:

$$\left(\|\,(\nabla F(\theta_t))_{active}\,\|\sqrt{d_{\text{active}}+2}+\|\,(\nabla F(\theta_t))_{\perp}\,\|\sqrt{d_{\perp}+2}\right)^2 < \|\nabla F(\theta_t)\|^2(d+2) \qquad (13)$$

Let $a_1 = \sqrt{d_{\text{active}}+2}, a_2 = \sqrt{d_{\perp}+2}, b_1 = \|\,(\nabla F(\theta_t))_{active}\,\|, b_2 = \|\,(\nabla F(\theta_t))_{\perp}\,\|, a = \sqrt{d+2}$ and $b = \|\nabla F(x_t)\|$.

The following relationships hold: $b_1^2 + b_2^2 = b^2$ and $a_1^2 + a_2^2 - 2 = a^2$. The bound we want to prove in Equation 13 reduces to finding conditions under which:

$$(a_1 b_1 + a_2 b_2)^2 \leq (b_1^2 + b_2^2)(a_1^2 + a_2^2 - 2)$$

Which holds iff:

$$2b_1^2 + 2b_2^2 \leq a_2^2 b_1^2 + a_1^2 b_2^2 - 2a_1 a_2 b_1 b_2$$

The later holds iff:

$$|a_1 b_2 - a_2 b_1| \geq \sqrt{2}b$$

Which holds iff:

$$\left|\sqrt{d_{\text{active}}+2}\|\,(\nabla F(x_t))_{\perp}\,\|-\sqrt{d_{\perp}+2}\|\,(\nabla F(x_t))_{active}\,\|\right| \geq \sqrt{2}\|\nabla F(x_t)\|$$

□

The inequality is strict for example when $\|\,(\nabla F(x_t))_{\perp}\,\| = 0$ and $d_{\perp} \geq 1$.

This in turn implies that, after taking into account the bias terms:

**Theorem 6.10.** *If $\epsilon > 0$. If $\sigma < \frac{1}{35}\sqrt{\frac{\epsilon \min((p^t)^*, 1-(p^t)^*)}{\tau d^3 \max(L,1)}}$, we denote by $Var_{(\widehat{P})^*}$ as the variance of the gradient estimator $z_1$ corresponding to the optimal (for $Var_{\widehat{P}}^M$) probability $(p^t)^*$ and*

$$|\sqrt{d_{\text{active}}+2}\|\,(\nabla F(\theta_t))_{\perp}\,\|-\sqrt{d_{\perp}+2}\|\nabla F(\theta_t)_{active}\||\geq \sqrt{2}\|\nabla F(\theta_t)\|$$

*Then:*

$$Var_{\widehat{P}^*} \leq Var_{ES} + \epsilon$$

## 6.3 Adaptive Mirror Descent for variance reduction.

In this section we propose an adaptive procedure to learn the optimal probability parameter $(p^t)^*$ (as introduced in the previous section) this is necessary since as it can be inferred from the discussion in section 6.2, the optimal variance depends of unknown parameters such as the projection of the true gradient onto the subspaces spanned by $\mathbf{U}$ and $\mathbf{U}^\perp$. The final result of this section 6.15 corresponds to Theorem 3.3 in the main section of the text.

Let $\mathbf{p}^l = \binom{p^l}{1-p^l}$. The main component $\Gamma$ of the variance of $\widehat{\nabla}_{\mathrm{MC},k=1}^{\mathrm{AT,asebo}} F_\sigma(\theta)$ as a function of $\mathbf{p}^l$ equals (Lemma 3.2) :

$$\Gamma = \ell(\mathbf{p}^l) = \frac{d_{\mathrm{active}} + 2}{\mathbf{p}^l(1)} s_{\mathbf{U}^{\mathrm{act}}} + \frac{d_\perp + 2}{\mathbf{p}^l(2)} s_{\mathbf{U}^\perp} - \|\nabla F(\theta)\|^2. \tag{14}$$

In order to avoid the gradients to explote in norm, we parametrise $\mathbf{l}^l$ as follows:

$$\mathbf{p}^l = (1 - 2\beta)\mathbf{q}^l + \binom{\beta}{\beta}$$

For $\mathbf{q}^l \in \Delta_2$ and $\beta \in (0,1)$, the boundary probability bias.

Notice that $\Gamma$ is a convex function of $\mathbf{p}$ and also a convex function of $\mathbf{q}$. With a slight abuse of notation we denote $\ell(\mathbf{q}^l)$ as the loss parametrized by $\mathbf{q}^l$ (which satisfies $\ell(\mathbf{q}^l) = \ell(\mathbf{p}^l)$).

The gradient $\nabla_{\mathbf{q}^l} \ell(\mathbf{q}^l)$ equals:

$$\nabla_{\mathbf{q}^l} \ell(\mathbf{q}^l) = (1 - 2\beta) \begin{pmatrix} -\frac{d_{\mathrm{active}}+2}{((1-2\beta)\mathbf{q}^l(1)+\beta)^2} s_{\mathbf{U}^{\mathrm{ort}}} \\ -\frac{d_\perp+2}{((1-2\beta)\mathbf{q}^l(2)+\beta)^2} s_{\mathbf{U}^\perp} \end{pmatrix},$$

And can be approximated (at the cost of some bias) using function evaluations.

**Lemma 6.11.** *The gradient $\nabla_{\mathbf{q}^l} \ell(\mathbf{q}^l)$ satisfies:*

$$\left\| \nabla_{\mathbf{q}^l} \ell(\mathbf{q}^l) - \mathbb{E}\left[ (1-2\beta)\begin{pmatrix} -\frac{a_l(d_{\mathrm{active}}+2)}{((1-2\beta)\mathbf{p}^l(1)+\beta)^3} \\ -\frac{(1-a_l)(d_\perp+2)}{((1-2\beta)\mathbf{p}^l(2)+\beta)^3} \end{pmatrix} v_l^2 \right] \right\| \leq \frac{\epsilon(d+2)}{(\min(\mathbf{p}^l(1),\mathbf{p}^l(2)))^3} \leq \frac{\epsilon(d+2)}{\beta^3},$$

*where $v_l = \frac{1}{2\sigma}\left( F(\theta + \mathbf{g}_l) - F(\theta - \mathbf{g}_l) \right).$*

*Proof.* We start with some notation borrowed from the previous section:

$$\xi_{\mathbf{g}_l}^{(2)}(\theta) = \underbrace{\left( \frac{F(\theta + \sigma \mathbf{g}_l) - F(\theta - \sigma \mathbf{g}_l)}{2\sigma} \right)^2}_{v_l^2} - \left( \mathbf{g}_l^\top \nabla F(\theta_t) \right)^2$$

Observe that:

$$|\xi_{\mathbf{g}_l}^{(2)}(\theta)| = \left| \left( \frac{F(\theta + \sigma \mathbf{g}_l) - F(\theta - \sigma \mathbf{g}_l)}{2\sigma} \right)^2 - \left( \mathbf{g}_l^\top \nabla F(\theta) \right)^2 \right|$$

$$\leq \xi_{\mathbf{g}_l}(\theta)^2 + 2\left| \mathbf{g}_l^\top \nabla F(\theta) \xi_{\mathbf{g}_l}(\theta) \right|$$

$$\leq \sigma^4 \tau^2 \|\mathbf{g}_l\|^6 + 2\sigma^2 \tau L \|\mathbf{g}_l\|^4$$

Since $\sigma < \frac{1}{35}\sqrt{\frac{\epsilon \min(p^t, 1-p^t)}{\tau d^3 \max(L,1)}}$:

$$\mathbb{E}\left[ |\xi_{\mathbf{g}_l}^{(2)}(\theta)| \right] \leq \epsilon$$

The result follows.

$\square$

Let $p^l = (1 - 2\beta)q^l + \beta$ be the probability that we choose to sample from the subspace $\mathcal{L}^{ES}_{active}$ and $1 - p^l$ the probability that we choose to sample from $\mathcal{L}^{ES,\perp}_{active}$. Let $a_l$ be a Bernoulli random variable $a_l \in \{0, 1\}$ with $\mathbb{E}\left[\binom{a_l}{1-a_l}\right] = \mathbf{p}^l$. Define the stochastic gradient (with respect to $\mathbf{q}^l$):

$$\mathbf{e}_l = (1 - 2\beta)\left[\begin{pmatrix} -\frac{a_l(d_{active}+2)}{p_l^3} \\ -\frac{(1-a_l)(d_\perp+2)}{(1-p_l)^3} \end{pmatrix}\right] v_l^2$$

By definition this random vector (conditioned on the choice of $\mathbf{p}^l$) satisfies:

$$\left\|\mathbb{E}\left[e_l\right] - \nabla_{\mathbf{q}^l}\ell(\mathbf{q}^l)\right\| \le \frac{\epsilon(d+2)}{(\min(\mathbf{p}^l(1), \mathbf{p}^l(2)))^3} \le \frac{\epsilon(d+2)}{\beta^3},$$

If $\epsilon$ is chosen small enough, the bias can be driven to be arbitrarily small.

### 6.3.1 Mirror descent

We treat this problem as that of minimizing the loss $\ell$ over the two dimensional simplex and resort to adapt a version of Mirror descent for it. As opposed the case of projected gradient descent, mirror descent performs updates that are adapted to the geometry of the simplex, ensuring the iterates always belong to the simplex and no projection step is necessary. The mirror descent updates are:

$$\mathbf{q}_l(1) = \frac{\mathbf{q}_{l-1}(1)\exp(-\alpha\mathbf{e}_l(1))}{\mathbf{q}_{l-1}(1)\exp(-\alpha\mathbf{e}_l(1)) + (\mathbf{q}_{l-1}(2))\exp(-\alpha\mathbf{e}_l(2))}$$
$$\mathbf{q}_l(2) = \frac{\mathbf{q}_{l-1}(2)\exp(-\alpha\mathbf{e}_l(2))}{\mathbf{q}_{l-1}(1)\exp(-\alpha\mathbf{e}_l(1)) + (\mathbf{q}_{l-1}(2))\exp(-\alpha\mathbf{e}_l(2))}$$

For a step size parameter $\alpha$.

## 6.4 Regret guarantees

Using he notation in https://www.stat.berkeley.edu/~bartlett/courses/2014fall-cs294stat260/lectures/mirror-descent-notes.pdf, In this case let $R(\mathbf{q}) = \mathbf{q}(1)\log(\mathbf{q}(1)) + \mathbf{q}(2)\log(\mathbf{q}(2)) - \mathbf{q}(1) - \mathbf{q}(2)$ and therefore:

$$\nabla R(\mathbf{q}) = \begin{pmatrix}\log(\mathbf{q}(1)) \\ \log(\mathbf{q}(2))\end{pmatrix} \tag{15}$$

The Fenchel conjugate of $R$ equals:

$$R^*(\mathbf{q}) = e^{\mathbf{q}(1)} + e^{\mathbf{q}(2)} \tag{16}$$

And therefore the gradient of the Fenchel conjugate equals:

$$\nabla R^*(\mathbf{q}) = \begin{pmatrix}\exp(\mathbf{q}(1)) \\ \exp(\mathbf{q}(2))\end{pmatrix} \tag{17}$$

And:

$$D_R(\mathbf{q}_1, \mathbf{q}_2) = \mathbf{q}_1(1)\log\left(\frac{\mathbf{q}_1(1)}{\mathbf{q}_2(1)}\right) + \mathbf{q}_1(2)\log\left(\frac{\mathbf{q}_1(2)}{\mathbf{q}_2(2)}\right) + \mathbf{q}_2(1) - \mathbf{q}_1(1) + \mathbf{q}_2(2) - \mathbf{q}_1(2) \tag{18}$$

Recall the update behind Mirror descent takes the form (stepsize $\alpha$:

1. Play $\binom{a_l}{1-a_l}$ such that $\mathbb{E}[\binom{a_l}{1-a_l}] = \mathbf{p}^l$.
2. Let $w_{l+1} = \nabla R^*\left(\nabla R(\mathbf{p}^l) - \alpha\mathbf{e}_l\right)$

3. Let $\mathbf{p}^{l+1} = \arg\min_{\mathbf{p} \in \Delta_2} D_R(\mathbf{p}, w_{t+1})$

Recall the general definition of Bregman divergence:

$$D_\Psi(u, v) = \Psi(u) - \Psi(v) - \langle \nabla\Psi(v), u - v \rangle \tag{19}$$

The following regret guarantee holds for Mirror descent (see https://www.stat.berkeley.edu/~bartlett/courses/2014fall-cs294stat260/lectures/mirror-descent-notes.pdf):

**Theorem 6.12.** *If at time $l$ a convex loss function $f_l$ is revealed to the player and the player performs the mirror descent step using $\nabla f_l$ as a proxy linear function, with actions (from the mirror descent step) $a_l$ at time $l$, for any $a$ in the intersection of all of $f_l$'s domains, the following regret bound holds:*

$$\sum_{l=1}^{C} (f_l(a_l) - f_l(a)) \leq \sum_{l=1}^{C} \nabla f_l(a_l)^\top (a_l - a) \tag{20}$$

$$\leq \frac{1}{\alpha}\left( R(a) - R(a_1) + \sum_{l=1}^{C} D_{R^*}(\nabla R(a_l) - \alpha \nabla f_l(a_l), \nabla R(a_l)) \right) \tag{21}$$

Also remember that if $R^*$ is $\theta$-smooth with respect to some norm $\|\cdot\|$, we can upper bound $D_{R^*}$. The former ($R^*$ being $\theta$-smooth) holds if $R$ is $\frac{1}{\theta}$-strongly convex with respect to the dual norm $\|\cdot\|_*$. When $R$ equals the entropy, this is $1$-strongly convex with respect to the $L_1$ norm and hence $R^*$ is $1$-strongly smooth with respect to the $L_\infty$ norm:

$$D_{R^*}(a, b) \leq \frac{\|a - b\|_\infty^2}{2} \tag{22}$$

In our case, let $f_l(\mathbf{q}) = \mathbf{e}_l^\top \mathbf{q}$. Using the upper bound previously described for $D_R$. For any $\mathbf{q} \in \Delta_2$:

$$\sum_{l=1}^{C} f_l(\mathbf{q}^l) - f_l(\mathbf{q}) \leq \frac{1}{\alpha}\left( R(\mathbf{q}) - R(\mathbf{q}^1) + \alpha^2 \sum_{l=1}^{C} \frac{\|\nabla f_l(\mathbf{q}_l)\|_\infty^2}{2} \right)$$

Taking expectations, since $\left| \mathbb{E}[f_l(\mathbf{q})|\mathbf{q}_l] - \nabla_{\mathbf{q}^l}^\top \ell(\mathbf{q}^l)\mathbf{q} \right| \leq \frac{\epsilon(d+2)}{\beta^3}$ we obtain the following result:

$$\left( \sum_{l=1}^{C} \nabla_{\mathbf{q}^l}^\top \ell(\mathbf{q}^l)(\mathbf{q}^l - \mathbf{q}) \right) - C\frac{2\epsilon(d+2)}{\beta^3} \leq \frac{1}{\alpha}\left( R(\mathbf{q}) - \mathbb{E}\left[R(\mathbf{q}^1)\right] + \alpha^2 \sum_{l=1}^{C} \frac{\mathbb{E}\left[\|\mathbf{e}_l\|_\infty^2\right]}{2} \right)$$

Now we bound the Right Hand side of the expression above. Notice that $R(\mathbf{q}) \leq 2$ and $R(\mathbf{q}_1) \geq 0$. We can also bound the expectation $\mathbb{E}\left[\|\mathbf{e}_l\|_\infty^2\right]$.

**Lemma 6.13.** $\left| \mathbb{E}\left[\|\mathbf{e}_l\|^2\right] - \|\nabla_{\mathbf{q}^l}\ell(\mathbf{q}^l)\|^2 \right| \leq \frac{\epsilon(d+1)}{\beta^3}$

*Proof.* A similar calculation as in Lemma 6.11 yields the desired result. $\square$

Since:

$$\mathbb{E}\left[\|\mathbf{e}_l\|_\infty^2\right] \leq \mathbb{E}\left[\|\mathbf{e}_l\|^2\right] \tag{23}$$

And $\|\nabla_{\mathbf{q}^l}\ell(\mathbf{q}^l)\|^2 \leq \frac{1}{\beta^4}\left( (d_{\text{active}} + 2)^2 s_{\mathbf{U}^{\text{ort}}}^2 + (d_\perp + 2)^2 s_{\mathbf{U}^\perp}^2 \right)$.

We obtain the following bound:

$$\left( \sum_{l=1}^{C} \nabla_{\mathbf{q}^l}^\top \ell(\mathbf{q}^l)(\mathbf{q}^l - \mathbf{q}) \right) - C\frac{2\epsilon(d+2)}{\beta^3} \leq \frac{2}{\alpha} + \frac{\alpha C}{2\beta^4}\left( (d_{\text{active}} + 2)^2 s_{\mathbf{U}^{\text{ort}}}^2 + (d_\perp + 2)^2 s_{\mathbf{U}^\perp}^2 \right) + \frac{\alpha C\epsilon(d+1)}{\beta^3}$$

The following theorem follows:

**Theorem 6.14.** *If* $\alpha = \frac{2\beta^2}{\sqrt{C}\sqrt{(d_{\text{active}}+2)^2 s_{\mathbf{U}^{\text{ort}}}^2 + (d_\perp+2)s_{\mathbf{U}^\perp}^2}}$ *and* $\epsilon = \frac{\beta^3}{2C(d+1)}$, *for any* $\mathbf{q} \in \Delta_2$:

$$\left(\sum_{l=1}^{C} \nabla_{\mathbf{q}^l}^\top \ell(\mathbf{q}^l) \left(\mathbf{q}^l - \mathbf{q}\right)\right) \leq \frac{\sqrt{C}\sqrt{(d_{\text{active}}+2)^2 s_{\mathbf{U}^{\text{ort}}}^2 + (d_\perp+2)s_{\mathbf{U}^\perp}^2}}{\beta^2} + 1$$

*Proof.* Plugging in this value of $\alpha$:

$$\left(\sum_{l=1}^{C} \nabla_{\mathbf{q}^l}^\top \ell(\mathbf{q}^l) \left(\mathbf{q}^l - \mathbf{q}\right)\right) \leq \frac{\sqrt{C}\sqrt{(d_{\text{active}}+2)^2 s_{\mathbf{U}^{\text{ort}}}^2 + (d_\perp+2)s_{\mathbf{U}^\perp}^2}}{\beta^2}$$
$$+ \left(1 + \frac{2\beta^2}{\sqrt{C}\sqrt{(d_{\text{active}}+2)^2 s_{\mathbf{U}^{\text{ort}}}^2 + (d_\perp+2)s_{\mathbf{U}^\perp}^2}}\right) \frac{C\epsilon(d+1)}{\beta^3}$$

By setting $\epsilon = \frac{\beta^3}{2C(d+1)}$ the result follows. Assuming $C$ is large enough so that $\alpha < 1$. $\qquad\square$

Since $\ell(\mathbf{q})$ is a convex function of $\mathbf{q}$ for all $l$ and $\mathbf{q} \in \Delta_2$:

$$\ell(\mathbf{q}^l) - \ell(\mathbf{q}) \leq \nabla_{\mathbf{q}^l}^\top \ell(\mathbf{q}^l) \left(\mathbf{q}^l - \mathbf{q}\right)$$

Which in turn implies the main result of this section:

**Theorem 6.15.** *If* $\alpha = \frac{2\beta^2}{\sqrt{C}\sqrt{(d_{\text{active}}+2)^2 s_{\mathbf{U}^{\text{ort}}}^2 + (d_\perp+2)s_{\mathbf{U}^\perp}^2}}$ *and* $\epsilon = \frac{\beta^3}{2C(d+1)}$, *for any* $\mathbf{q} \in \Delta_2$:

$$\mathbb{E}\left[\sum_{l=1}^{C} \ell(\mathbf{q}^l) - \ell(\mathbf{q})\right] \leq \frac{\sqrt{C}\sqrt{(d_{\text{active}}+2)^2 s_{\mathbf{U}^{\text{ort}}}^2 + (d_\perp+2)s_{\mathbf{U}^\perp}^2}}{\beta^2} + 1$$

*This is equivalent to the result stated in the main paper.*

# 7 Additional Implementation Details

In this section we present additional details on our experimental results, for both the RL tasks and Nevergrad functions.

## 7.1 Reinforcement Learning Experiment Details

We provide additional details regarding the RL experiments below.

**State Normalization.** State-of-the-art policy optimization baselines such as PPO/TRPO [12] and the original ARS [25] apply state normalization as part of the implementation. In particular, the algorithms maintain a component-wise running average of mean $\bar{s}$ and standard deviation vector $\sigma(s)$ of the state. When at given state $s_t$, the algorithm computes the normalized state $\tilde{s}_t = \frac{s_t - \bar{s}}{\sigma(s)}$ before inputing to the policy network to compute actions $a_t = \pi(\tilde{s}_t)$. For PPO/TRPO, since the optimization is based on back-propagation of neural networks, properly scaling the inputs $s_t \rightarrow \tilde{s}_t$ is critical for the performance. In all experiments, we remove state normalization mechanism from the implementation to test the robustness of various blackbox optimization algorithms. Notice that as reported by [25], state normalization was not needed in ARS to learn good policies for RL tasks under consideration in this paper. As a result, we observe that PPO/TRPO underperform other ES algorithms for most tasks.

**Benchmark Environments.** Benchmark environments are from OpenAI gym [6]. These environments have variable sizes of observation space and action space: Swimmer-v2 $|\mathcal{S}| = 8$, $|\mathcal{A}| = 2$; Hopper-v2 $|\mathcal{S}| = 11$, $|\mathcal{A}| = 3$; HalfCheetah-v2 $|\mathcal{S}| = 17$, $|\mathcal{A}| = 6$; Thrower-v2 $|\mathcal{S}| = 23$, $|\mathcal{A}| = 7$; Pusher-v2 $|\mathcal{S}| = 23$, $|\mathcal{A}| = 7$; Walker2d-v2 $|\mathcal{S}| = 17$, $|\mathcal{A}| = 6$; Reacher-v2 $|\mathcal{S}| = 11$, $|\mathcal{A}| = 2$. All environments have a natural termination condition specified in the simulation environment.

**Policy Architecture.** All baseline algorithms involve training a parameterized policy $\pi_\theta(a|s)$ using sample gradient estimates generated from the environment. The policy architecture is shared across all algorithms: a 2-layer feed-forward neural network with tanh non-linearity and $h$ hidden units per layer. The input to the network is the state $s \in \mathcal{S}$. For all ES-based algorithms (Vanilla ES, CMA-ES, ARS and ASEBO), the output of the network is the action $a_\theta(s) \in \mathcal{A}$. For policy optimization algorithms (PPO, TRPO), the output of the network is a mean of Gaussian $\mu_\theta(s)$ and we draw actions from a factorized Gaussian distribution $a \sim \mathcal{N}(\mu_\theta(s), \sigma^2 \mathbb{I})$ where we separately parameterize a standard deviation parameter $\sigma$ shared across dimensions. The sizes of hidden layers where: 4 for LQR, 16 for Swimmer-v2, Hopper-v2 and Reacher-v2, 32 for HalfCheetah-v2 and Walker2d-v2, reflecting the difficulty of each task.

**Optimization.** Our method (ASEBO) and most baselines (Vanilla ES, ARS, CMA-ES and PPO) apply SGD based methods and we apply the Adam optimizer to stabilize the gradients.

### 7.1.1 Baseline Algorithms

**Vanilla ES.** Vanilla ES is the simplest evolutionary algorithm applied in RL tasks [7, 30]. We apply the antithetic sampling scheme as applied in [7]. Our implementation does not rank the rewards as in [30], and as previously discussed does not include observation normalization.

**CMA-ES variants.** Covariance Matrix Adaptation Evolution Strategy is a state-of-the-art and popular black box optimization algorithm [16]. VkD-CMA-ES and LM-MA-ES are recently proposed variant designed for high dimensional blackbox functions. For VkD-CMA-ES we use the open source implementation from *pycma* available at `http://github.com/CMA-ES/pycma`. We use the default hyper-parameters in the original code base with the standard deviation parameter $\sigma = 1.0$. For LM-MA-ES we use the implementation from [26].

**ARS.** Augmented Random Search [25] is based on the code released by the original paper. We use the standard deviation $\sigma = 0.02$ and learning rate $\eta = 0.01$. The hyper-parameters are tuned on top of the default hyper-parameters in the original code base. We remove the observation normalization utility in the original code for fair comparison.

**ASEBO.** We propose Adaptive Sample Efficient Blackbox Optimization in this work. Our algorithms have the following hyper-parameters: the covariance decay parameter $\lambda = 0.995$ (slow decay), proportion of variance of the active (PCA) space $\epsilon = 0.995$, standard deviation parameter $\sigma = 0.02$. We set the learning rate $\eta = 0.02$.

**Trust Region Policy Optimization.** Trust Region Policy Optimization (TRPO) is based on the implementation of OpenAI baseline [12]. We use the default training hyper-parameters in the code base: we collect $N = 1024$ samples per batch to compute a policy gradient, with the trust region size parameter $\epsilon = 0.01$. We remove the observation normalization utility in the original code for fair comparison.

**Proximal Policy Optimization.** Proximal Policy Optimization (PPO) [32] is also based on the implementation of OpenAI baseline [12]. We use the default hyper-parameters in the code base: we collect $N = 2048$ samples per batch to compute policy gradients and set the clipping coefficient $\epsilon = 0.2$. The learning rate is set to be $\alpha = 3 \cdot 10^{-5}$ for all environments. We remove the observation normalization utility in the original code for fair comparison.

## 7.2 Nevergrad **Experiment Details**

**Function Settings.** We tested the following functions: cigar, ellipsoid, sphere, sphere2, rosenbrock, rastrigin and lunacek. In each case we used $d = 1000$ to evaluate ASEBO in a high dimensional setting.

**Algorithm Hyper-Parameters.** We use the same hyper-parameters across all functions. For ASEBO and VanillaES, we use $\eta = 0.02$. For ASEBO we set $\lambda = 0.99$. For VkD-CMA-ES we use the default parameters from the pycma package, and for LM-MA-ES we use the implementation from [26].

## 8 The Algorithm - Additional Details & Analysis

We provide here few variations of the ASEBO algorithm from the main body of the paper, namely:

---

**Algorithm 3** ASEBO Algorithm - extended version

---

**Hyperparameters:** number of iterations of full sampling $l$, smoothing parameter $\sigma > 0$, step size $\eta$, PCA threshold $\epsilon$, decay rate $\gamma$, total number of iterations $T$.
**Input:** blackbox function $F$, vector $\theta_0 \in \mathbb{R}^d$ where optimization starts. $\text{Cov}_0 \in \{0\}^{d \times d}$, $p^0 = 0$.
**Output:** vector $\theta_T$.
**for** $t = 0, \ldots, T - 1$ **do**
    **if** $t < l$ **then**
        Take $n_t = d$. Sample $\mathbf{g}_1, \cdots, \mathbf{g}_{n_t}$ from $\mathcal{N}(0, \mathbf{I}_d)$ (independently).
    **else**
        1. Take top $r$ eigenvalues $\lambda_i$ of $\text{Cov}_t$, where $r$ is smallest such that: $\sum_{i=1}^{r} \lambda_i \geq \epsilon \sum_{i=1}^{d} \lambda_i$, using its SVD as described in text and take $n_t = r$.
        2. Take the corresponding eigenvectors $\mathbf{u}_1, ..., \mathbf{u}_r \in \mathbb{R}^d$ and let $\mathbf{U} \in \mathbb{R}^{d \times r}$ be obtained by stacking them together. Let $\mathbf{U}^{\text{act}} \in \mathbb{R}^{d \times r}$ be obtained from stacking together some orthonormal basis of $\mathcal{L}^{\text{ES}}_{\text{active}} \stackrel{\text{def}}{=} \text{span}\{\mathbf{u}_1, ..., \mathbf{u}_r\}$. Let $\mathbf{U}^{\perp} \in \mathbb{R}^{d \times (d-r)}$ be obtained from stacking together some orthonormal basis of the orthogonal complement $\mathcal{L}^{\text{ES},\perp}_{\text{active}}$ of $\mathcal{L}^{\text{ES}}_{\text{active}}$.
        3. Sample $\mathbf{g}_1, ..., \mathbf{g}_{n_t}$ from $\mathcal{N}(0, \sigma\Sigma)$ (independently), where $\Sigma = \frac{1-p^t}{d}\mathbf{I}_d + \frac{p^t}{r}\mathbf{U}\mathbf{U}^{\top}$ **(V0)** or sample $n_t$ vectors $\mathbf{g}_1, ..., \mathbf{g}_{n_t}$ as follows: with probability $1 - p^t$ from $\mathcal{N}(0, \mathbf{U}^{\perp}(\mathbf{U}^{\perp})^{\top})$ and with probability $p^t$ from $\mathcal{N}(0, \mathbf{U}^{\text{act}}(\mathbf{U}^{\text{act}})^{\top})$ **(V1)**.
        4. Renormalize $\mathbf{g}_1, ..., \mathbf{g}_{n_t}$ such that marginal distributions $\|\mathbf{g}_i\|_2$ are $\chi(d)$.
    1. Compute $\widehat{\nabla}^{\text{AT}}_{\text{MC}} F(\theta_t)$ as:

$$\widehat{\nabla}^{\text{AT}}_{\text{MC}} F(\theta_t) = \frac{1}{2n_t \sigma} \sum_{j=1}^{n_t} (F(\theta_t + \mathbf{g}_j) - F(\theta_t - \mathbf{g}_j))\mathbf{g}_j.$$

    2. Set $\text{Cov}_{t+1} = \lambda\text{Cov}_t + (1 - \lambda)\Gamma$, where $\Gamma = \widehat{\nabla}^{\text{AT}}_{\text{MC}} F_\sigma(\theta_t)(\widehat{\nabla}^{\text{AT}}_{\text{MC}} F_\sigma(\theta_t))^{\top}$.
    3. Set $p^{t+1} = p_{\text{opt}}$ for $p_{\text{opt}}$ output by Algorithm 2 (from the main body) or $p^{t+1} = \frac{\widehat{r}}{\widehat{r}+1}$, where:

$$\widehat{r} = \frac{\|(\widehat{\nabla} F_\sigma(\theta_t))_{\text{active}}\|_2}{\|(\widehat{\nabla} F_\sigma(\theta_t))_{\perp}\|_2},$$

    is computed by Algorithm 4 (see: below) and scalars $\|(\widehat{\nabla} F_\sigma(\theta_t))_{\text{active}}\|_2$, $\|(\widehat{\nabla} F_\sigma(\theta_t))_{\perp}\|_2$ stand for the estimates of $\|(\nabla F_\sigma(\theta_t))_{\text{active}}\|_2$ and $\|(\nabla F_\sigma(\theta_t))_{\perp}\|_2$.
    4. Set $\theta_{t+1} \leftarrow \theta_t + \eta\widehat{\nabla}^{\text{AT}}_{\text{MC}} F(\theta_t)$.

---

- we propose one more method for sampling from heterogeneous distributions (see: version **V0** in Algorithm 3; the default one that we present in the main body is called **V1** here),

- we propose to use compressed sensing techniques (Algorithm 4) as an alternative to the contextual bandits method from the main body (Algorithm 2); the bandits method can be seen as an extension of the compressed sensing techniques.

---

**Algorithm 4** Explore estimator via compressed sensing

---

**Hyperparameters:** smoothing parameter $\sigma$, horizon $C$.
**Input:** subspaces: $\mathcal{L}_{\text{active}}^{\text{ES}}$, $\mathcal{L}_{\text{active}}^{\text{ES},\perp}$, function $F$, vector $\theta_t$.
**Output:** ratio $\widehat{r}$.
1. Initialize square norm averages $s_0^{\text{active}} = s_0^{\perp} = 0$.
**for** $l = 1, \cdots, C$ **do**

> 1. Sample $\mathbf{g}_l^{\text{active}} \sim \mathcal{N}(0, \sigma \mathbf{I}_{\mathcal{L}_{\text{active}}^{\text{ES}}})$.
> 2. Sample $\mathbf{g}_l^{\perp} \sim \mathcal{N}(0, \sigma \mathbf{I}_{\mathcal{L}_{\text{active}}^{\text{ES},\perp}})$.
> 3. Ask for $F(\theta_t \pm \mathbf{g}_l^{\text{type}})$ for type $\in \{\text{active}, \perp\}$.
> 4. Compute $v_l^{\text{type}} = \frac{1}{2\sigma}(F(\theta_t + \mathbf{g}_l^{\text{type}}) - F(\theta_t - \mathbf{g}_l^{\text{type}}))$.
> 5. Compute $s_l^{\text{active}} = \frac{l-1}{l} * s_{l-1}^{\text{active}} + \frac{(v_l^{\text{active}})^2}{l}$.
> 6. Compute $s_l^{\perp} = \frac{l-1}{l} * s_{l-1}^{\perp} + \frac{(v_l^{\perp})^2}{l}$.

**Return:** $\widehat{r} = \sqrt{\frac{s_C^{\text{active}}}{s_C^{\perp}}}$.

---

## 8.1 Estimating the sensing ratio $r$.

In this section we provide guarantees for the estimation of the ratio $r$ as specified in Section 2.3 for Algorithm 4. Recall the definitions $s_{\mathbf{U}^{\text{act}}} = \|\mathbf{U}^{\top}\nabla F(\theta_t)\|^2$ and $s_{\mathbf{U}^{\perp}} = \|(\mathbf{U}^{\perp})^{\top}\nabla F(\theta_t)\|^2$.

Since $\left| \frac{F(\theta_t + \sigma \mathbf{g}) - F(\theta_t - \sigma \mathbf{g})}{2\sigma} - \mathbf{g}^{\top}\nabla F(\theta_t) \right| \leq \xi_{\mathbf{g}}(\theta_t)$, when $\mathbf{g} \sim \widehat{P}$, we recognize two cases. If $\mathbf{g} \sim \mathcal{N}(0, \mathbf{I}_{\mathbf{U}})$ the distribution of $\frac{F(\theta_t + \sigma \mathbf{g}) - F(\theta_t - \sigma \mathbf{g})}{2\sigma} \approx N(0, \|\mathbf{U}^{\top}\nabla F(\theta_t)\|^2)$. Analogously when $\mathbf{g} \sim \mathcal{N}(0, \mathbf{I}_{\mathbf{U}^{\perp}})$ the distribution of $\frac{F(\theta_t + \sigma \mathbf{g}) - F(\theta_t - \sigma \mathbf{g})}{2\sigma} \approx N(0, \|(\mathbf{U}^{\perp})^{\top}\nabla F(\theta_t)\|^2)$.

**Theorem 8.1.** *Let $0 < s < C$ and let $\mathbf{g}_i \sim \mathcal{N}(0, \mathbf{I}_{\mathcal{L}_{(\text{active})}^{\text{ES}}})$ for $i = 1, ..., s$ and $\mathbf{g}_i, \sim \mathcal{N}(0, \mathbf{I}_{\mathcal{L}_{\text{active}}^{\text{ES},\perp}})$ for $i = s+1, ..., C$. Let $\hat{s}_{\mathbf{U}^{\text{ort}}} := \frac{1}{s}\sum_{j=1}^{s}\left(\frac{F(\theta + \sigma \mathbf{g}_j) - F(\theta - \sigma \mathbf{g}_j)}{2\sigma}\right)^2$, $\hat{s}_{\mathbf{U}^{\perp}} := \frac{1}{C-s}\sum_{j=1}^{C-s}\left(\frac{F(\theta + \sigma \mathbf{g}_j) - F(\theta - \sigma \mathbf{g}_j)}{2\sigma}\right)^2$ and let $\hat{r} = \sqrt{\frac{\hat{s}_{\mathbf{U}^{\text{ort}}}}{\hat{s}_{\mathbf{U}^{\perp}}}}$. Given $u, \epsilon > 0$ and $\delta \in (\epsilon, 1)$, the following holds.*

> 1. *If $C = 2s$ for $s \geq \frac{16}{u^2}\log\left(\frac{8}{\delta}\right)$ and under the mechanism from Algorithm 4 or*

> 2. *If $\{\mathbf{g}_i\}_{i=1}^{C}$ are samples generated under $\widehat{P}$, $\min(p^t, 1 - p^t) > u$ and $C \geq \max\left(\frac{8}{(p^t - u)u^2}, \frac{8}{(1 - p^t - u)u^2}, \frac{2p^t + 2u/3}{u^2}\right)\log\left(\frac{12}{\delta}\right)$,*

*then with probability at least $1 - \delta$:*

$$\sqrt{\frac{s_{\mathbf{U}^{\text{ort}}}(1 - u) - \frac{2\epsilon}{\delta}}{s_{\mathbf{U}^{\perp}}(1 + u) + \frac{2\epsilon}{\delta}}} \leq \widehat{r} \leq \sqrt{\frac{s_{\mathbf{U}^{\text{act}}}(1 + u) + \frac{2\epsilon}{\delta}}{s_{\mathbf{U}^{\perp}}(1 - u) - \frac{2\epsilon}{\delta}}}.$$

*Proof.* First observe we introduce some notation.

$$\xi_{\mathbf{g}}^{(2)}(\theta_t) = \left(\frac{F(\theta_t + \sigma \mathbf{g}) - F(\theta_t - \sigma \mathbf{g})}{2\sigma}\right)^2 - \left(\mathbf{g}^{\top}\nabla F(\theta_t)\right)^2$$

Observe that:

$$\xi_{\mathbf{g}}^{(2)}(\theta_t) = \left| \left(\frac{F(\theta_t + \sigma \mathbf{g}) - F(\theta_t - \sigma \mathbf{g})}{2\sigma}\right)^2 - \left(\mathbf{g}^{\top}\nabla F(\theta_t)\right)^2 \right|$$

$$\leq \xi_{\mathbf{g}}(\theta_t)^2 + 2\left|\mathbf{g}^{\top}\nabla F(\theta_t)\xi_{\mathbf{g}}(\theta_t)\right|$$

$$\leq \sigma^4 \tau^2 \|\mathbf{g}\|^6 + 2\sigma^2 \tau L \|\mathbf{g}\|^4$$

Let $\hat{s}_V = \hat{s}_V^0 + \frac{1}{s}\sum_{j=1}^{s}\xi_{\mathbf{g}_j}(\theta_t)$ and $\hat{s}_{V\perp} = \hat{s}_{V\perp}^0 + \frac{1}{C-s}\sum_{j=1}^{C-s}\xi_{\mathbf{g}_j}(\theta_t)$. Where $s_{\mathbf{U}^{\mathrm{act}}}^0 = \frac{1}{s}\sum_{j=1}^{s}\left(\nabla F(\theta_t)^\top \mathbf{g}_j\right)^2$ and $s_{\mathbf{U}^\perp}^0 = \frac{1}{C-s}\sum_{j=s}^{C}\left(\nabla F(\theta_t)^\top \mathbf{g}_j\right)^2$.

Notice that $\nabla F(\theta_t)^\top \mathbf{g}$ is distributed as a Gaussian Random variable (with variance depending on the support of the covariance of $\mathbf{g}$).

By concentration of squared gaussian random variables:

$$\mathbb{P}\left[|\hat{s}_V^0 - s_{\mathbf{U}^{\mathrm{act}}}| \geq u s_{\mathbf{U}^{\mathrm{act}}}\right] \leq 2\exp\left(-\frac{su^2}{8}\right)$$

$$\mathbb{P}\left[|\hat{s}_{V\perp}^0 - s_{\mathbf{U}^\perp}| \geq u s_{\mathbf{U}^\perp}\right] \leq 2\exp\left(-\frac{(k-s)u^2}{8}\right)$$

Consequently, with probability $1 - 2\exp\left(-\frac{su^2}{8}\right) - 2\exp\left(-\frac{(k-s)u^2}{8}\right)$, it holds that:

$$\frac{s_{\mathbf{U}^{\mathrm{act}}}}{s_{\mathbf{U}^\perp}}\left(\frac{1+u}{1-u}\right) \geq \frac{\hat{s}_V^0}{\hat{s}_{V\perp}^0} \geq \frac{s_{\mathbf{U}^{\mathrm{act}}}}{s_{\mathbf{U}^\perp}}\left(\frac{1-u}{1+u}\right)$$

Notice that by Markov's inequality:

$$\mathbb{P}\left(\xi_{\mathbf{g}}(\theta_t) \geq \frac{2\epsilon}{\delta}\right) \leq \delta\frac{\mathbb{E}\left[\sigma^4\tau^2\|\mathbf{g}\|^6 + 2\sigma^2\tau L\|\mathbf{g}\|^4\right]}{2\epsilon} \leq \frac{\delta}{2} \tag{24}$$

Since $\sigma < \frac{1}{35}\sqrt{\frac{\epsilon\min(p^t, 1-p^t)}{\tau d^3\max(L,1)}}$, $\mathbb{E}\left[\sigma^4\tau^2\|\mathbf{g}\|^6 + 2\sigma^2\tau L\|\mathbf{g}\|^4\right] \leq \epsilon$.

Regardless if $\mathbf{g}$ was sampled from $\mathcal{N}(0,\mathbf{I})$, $\mathcal{N}(0,\mathbf{I}_\mathbf{U})$ or $\mathcal{N}(0,\mathbf{I}_{\mathbf{U}^\perp})$.

**Case 1.**

By definition, $C = 2s$, and therefore $C - s = C/2$ and therefore:

$$\mathbb{P}\left[|\hat{s}_V^0 - s_{\mathbf{U}^{\mathrm{act}}}| \geq u s_{\mathbf{U}^{\mathrm{act}}}\right] \leq 2\exp\left(-\frac{Cu^2}{16}\right)$$

$$\mathbb{P}\left[|\hat{s}_{V\perp}^0 - s_{\mathbf{U}^\perp}| \geq u s_{\mathbf{U}^\perp}\right] \leq 2\exp\left(-\frac{Cu^2}{16}\right)$$

We require that:

$$2\exp\left(-\frac{Cu^2}{16}\right) \leq \delta/4$$

$$2\exp\left(-\frac{Cu^2}{16}\right) \leq \delta/4$$

**Case 2.**

In fact, by concentration results on Bernoulli variables, given $\alpha > 0$, $|s - kp^t| \leq k\alpha$ and $|(k-s) - (1-p^t)k| \leq k\alpha$ with probability at least $1 - \exp\left(-\frac{k\alpha^2}{2p^t}\right) - \exp\left(-\frac{k\alpha^2}{2p^t+2\alpha/3}\right)$.

Let $\alpha = u$. Conditioning on the events that $|s - kp^t| \leq uk$ and $|(k-s) - (1-p^t)k| \leq uk$. We seek to ensure that:

$$2\exp\left(-\frac{(p^t-u)ku^2}{8}\right) \leq \delta/6$$

$$2\exp\left(-\frac{(1-p^t-u)ku^2}{8}\right) \leq \delta/6$$

$$\exp\left(-\frac{ku^2}{2p^t}\right) \leq \delta/6$$

$$\exp\left(-\frac{ku^2}{2p^t+2u/3}\right) \leq \delta/6$$

**Case 1 and 2** The following inequalities hold:

$$\frac{s_{\mathbf{U}^{\text{act}}}(1-u)-\frac{2\epsilon}{\delta}}{s_{\mathbf{U}^{\perp}}(1+u)+\frac{2\epsilon}{\delta}} \leq \frac{\hat{s}_V^0-\frac{2\epsilon}{\delta}}{\hat{s}_{V^{\perp}}^0+\frac{2\epsilon}{\delta}} \leq \frac{\hat{s}_V}{\hat{s}_{V^{\perp}}} \leq \frac{\hat{s}_V^0+\frac{2\epsilon}{\delta}}{\hat{s}_{V^{\perp}}^0-\frac{2\epsilon}{\delta}} \leq \frac{s_{\mathbf{U}^{\text{act}}}(1+u)+\frac{2\epsilon}{\delta}}{s_{\mathbf{U}^{\perp}}(1-u)-\frac{2\epsilon}{\delta}}$$

The union bound yields the desired result. And therefore the result follows. $\square$