[Reviews · NeurIPS 2019]

Reviewer 1



EDIT: I have read the author response; and appreciate the proposed improvements for the final version. I do not consider it necessary to change my score, or the review. The paper provides a new evolutionary strategy (ASEBO) for black-box optimization. The authors point out that ES methods scale poorly with dimensionality by trying to accurately estimate the gradient of the objective. Their method uses online PCA to estimate a subspace of the gradients, whose dimensionality varies during optimization. A factor controlling exploration is estimated using the gradient norm on the subspace and its complement. Experiments with RL tasks and synthetic functions show better performance of ASEBO compared to other ES methods, and commonly used algorithms. Originality / Quality: The paper presents a novel algorithm for black-box optimization. Both theoretical and experimental results are provided to demonstrate its effectiveness. The experiments are sound, and the authors compare to a number of popular methods in the domain. Clarity: Overall, the paper is well written and easy to follow. The theoretical results, and their significance, need to be discussed in more detail. This is especially true for Theorem 3.3. Significance: Given the renewed interest in black-box optimization, and the popularity of ES methods, I believe this work is an important contribution.

Reviewer 2



It is to be noted that the algorithm does not assume that the function is differentiable, but instead relies on Gaussian smoothing of the original function. Typical blackbox strategies evaluate the gradient of the Gaussian smoothing via MC techniques, then use typical first order optimization. However, sampling these gradients in high dimension is quite costly, which the article proposes to improve. To do so, it makes the assumption (validated by Fig1) that the gradients along the optimization process lie on a lower dimension subspace, the "active subspace", paving the way for cheaper evaluation. A first idea is to estimate these subspaces via PCA of the last s (parameter) estimated gradients, but s and the dimension of the PCA are difficult to tune. Instead, ASEBO uses bandit techniques to balance exploitation (sampling gradient along the active subspace) and exploration (sampling orthogonal to this set, via compressed sensing). The proposed methodology is sound, theoretical guarantees are provided and experiments seem to show that ASEBO needs fewer function evaluations than other existing approaches. Overall, the paper is easy to read even with mild knowledge about bandit techniques. A few points can be improved: The formulas in Algorithm 2 are not very intuitive and could benefit from better explanations. Several expressions are made clear long after they have been used for the first time: - The meaning "learn the bias of the low dimensional model" (L53) is only explained in 2 and was not clear to me before. - Similarly, the "hardness of the optimization problem" is not made clear from the beginning. - active subspace is explained L147 only. - When you speak about covariance matrix, you are speaking about the covariance of the gradients, which was not obvious to me until ~ L131. These could be put in Definition environments, and links to these defs given the first time the expression is mentioned, like "See formal definition in Def X" Cosmetic remarks: L131 you use lambda for decay rate, but gamma in Algorithm 1 hyperparams (better IMO since lambda is already taken for the eigenvalues of Cov_t) Technically lambda_i should be lambda_i,t since they depend on cov_t (nitpick but might be more explicit. otherwise r does not depend on t) L210 epsilon clashes with the epsilon of PCA used in Algo 1.

Reviewer 3



This work primarily introduces the methods of adaptive subspaces to the standard continuous control benchmarks through a BBO scheme. This introduction is particularly useful as these search methods have been proven competent in this setting; more efficient methods can open up new doors in related fields such as meta-learning, etc. The authors are through in their introduction of the method as well as their theoretical discussions and clearly discuss the active subspace methods, accurately relating it to previous work.

[Author Response · NeurIPS 2019]

We would like to sincerely thank all the reviewers for their very valuable feedback. Below we address reviewers' comments in more detail.

**Overview:** We are happy that the reviewers appreciate the importance of adaptive methods in ES and point out lots of applications and possible extensions of the presented results. We will discuss them as well as some of our own extension ideas in the final version.

**Additional Experiments:** In the final version we will include additional suggested experiments for linear policies and on RL tasks with no termination condition, that as noted by reviewers, might be particularly amenable to adaptive methods. Additionally, we agree that it would be very interesting to test ASEBO on even higher dimensional tasks such as Humanoid. This is well aligned with very recent research on intrinsic dimensionality for RL objective landscapes suggesting that tasks such as Humanoid exhibit much lower intrinsic dimensionality [1]. It would be useful to see how ASEBO can take advantage of this and thus we will include the results of these experiments in the final version.

**Theory Clarification:** We will also add a paragraph summarizing our theoretical findings. In particular, we will emphasize the adaptivity aspects (e.g. non-sensitivity to fixed hyperparameters). We will comment on Theorem 3.3 explaining that it shows the presented algorithm automatically finds the optimal (in terms of variance reduction) strategy of sampling from ES-active subspaces without any pre-tuned hyperparameters. In Theorem 3.3 this is expressed via convergence result through loss function $l$. We will clarify it.

**Open Sourcing:** Given the potential impact of adaptive methods to scale ES algorithms, we will release an open-source version of the algorithm with the final version of the paper (that will also contain an updated set of benchmark tests). We hope this will help other researchers in applying our algorithms for their RL problems, as well as developing them further.

**Additional Clarifications:**

- The curve for $n = 100$ is indeed hidden behind $n = 212$ in Fig.3 (as both failed to learn). Thank you for pointing this out. We will make this clearer in the final version.
- In the final version we will also simplify algorithmic block 2, as suggested and give an additional explanation in a separate paragraph.
- We will also add extra definitions and explanations of the used concepts that will be presented at the beginning of the paper, as recommended, to improve clarity.
- We will replace one of the plots in Fig.1 with the plot visualizing how the gradient direction changes as training progresses.
- We will also fix all listed typos regarding notations.

# References

[1] C. Li, H. Farkhoor, R. Liu, and J. Yosinski. Measuring the intrinsic dimension of objective landscapes. In *International Conference on Learning Representations*, 2018.


[Meta-Review · NeurIPS 2019]

All reviewers are positive about the paper. The paper addresses the problem of black-box optimization, currently of wide interest especially for reinforcement learning. The authors propose adaptive active subspaces techniques for black-box optimization. While the theoretical results seem currently limited, the experimental comparison is detailed and extensive. The proposed approach is therefore quite promising. We recommend to take the reviewers' comments and suggestions into account while preparing the camera ready final version of the paper. Accept.